



# Evaluation of a field-deployable Nafion-based air drying system for collecting whole air samples and its application to stable isotope measurements of $CO_2$

Dipayan Paul[1,*], Hubertus A. Scheeren[1,*], Henk G. Jansen[1], Bert A. M. Kers[1], John B. Miller[2], Andrew M. Crotwell[2,3], Sylvia E. Michel[4], Luciana V. Gatti[5], Lucas G. Domingues[5], Caio S. C. Correia[5], Raiane A. L. Neves[5], Harro A. J. Meijer[1], Wouter Peters[1,6]

[1]Centre for Isotope Research (CIO), University of Groningen, Groningen, the Netherlands
[2]National Oceanic and Atmospheric Administration (NOAA), Earth System Research Laboratory, Boulder, Colorado, USA
[3]Cooperative Institute for Research in Environmental Sciences (CIRES), University of Colorado, Boulder, Colorado, USA
[4]Institute of Arctic and Alpine Research (INSTAAR), University of Colorado, Boulder, Colorado, USA
[5]National Institute of Space research (INPE), Atmospheric Chemistry Department, San Jose dos Campos, Brazil
[6]Department of Meteorology and Air Quality, Environmental Sciences Group, Wageningen University, the Netherland

*both authors contributed equally.

Correspondence to: Dipayan Paul (d.paul@rug.nl) and Hubertus A. Scheeren (h.a.scheeren@rug.nl)

**Abstract.** Atmospheric flask samples are either collected at atmospheric pressure by simply opening a valve of a pre-evacuated flask, or pressurized with the help of a pump to a few bar above ambient providing large air samples for analysis. Under humid conditions, there is a risk that water vapour in the sample leads to condensation on the walls of the flask, notably at higher than ambient sampling pressures. Liquid water in sample flasks is known to affect the $CO_2$ mixing ratios and also alters the isotopic composition of oxygen ($^{17}O$ and $^{18}O$) in $CO_2$ via isotopic equilibration. Hence, for accurate determination of $CO_2$ mole fractions and its stable isotopic composition, it is vital to dry the air samples to a sufficiently low dew point before they are pressurized in flasks to avoid condensation. Moreover, the drying system itself should not influence the mixing ratio and the isotopic composition of $CO_2$, nor of the other constituents under study. For the "Airborne Stable Isotopes of Carbon from the Amazon" (ASICA) project focusing on accurate measurements of $CO_2$ and its singly-substituted stable isotopologues over the Amazon, an air drying system was needed capable of removing water vapour from air sampled at a dew point better than −2 °C, high flow rates up to 12 L/min, and without the need for electrical power. Since to date, no commercial air drying device is available that meets these requirements, we designed and built our own consumable-free, power-free, and portable drying system based on multi-tube Nafion™ gas sample driers (Perma Pure, Lakewood, USA). The required dry purge air is provided by feeding the exhaust flow of the flasks sampling system through a dry molecular sieve (type 3A) cartridge. In this study we describe the systematic evaluation of our Nafion-based air sample dryer with emphasis on its performance concerning the measurements of atmospheric $CO_2$ mole fractions and the three singly-substituted isotopologues of $CO_2$ ($^{16}O^{13}C^{16}O$, $^{16}O^{12}C^{17}O$ and $^{16}O^{12}C^{18}O$), as well as the trace gas species $CH_4$, $CO$,



$N_2O$, and $SF_6$. Experimental results simulating extreme tropical conditions (saturated air at 33 °C) indicated that the response of the air dryer is almost instantaneous and that approximately 85 L of air, containing up to 4% water vapour, can be

processed staying below a −2 °C dew point temperature (at 275 kPa). We estimated that least 8 flasks can be sampled (at an overpressure of 275 kPa) with a water vapour content below −2 °C dew point temperature during a typical flight sampling up to 5 km altitude over the Amazon, whereas the remaining samples would stay well below 5 °C dew point temperature (at 275 kPa). The performance of the air dryer on measurements of $CO_2$, $CH_4$, CO, $N_2O$, and $SF_6$, and the $CO_2$ isotopologues $^{16}O^{13}C^{16}O$ and $^{16}O^{12}C^{18}O$ was tested in the laboratory simulating real sampling conditions by compressing humidified air

from a calibrated cylinder, after being dried by the air dryer, into sample flasks. We found that the mole fraction and the isotopic composition difference between the different test conditions (including the dryer) and the base condition (dry air, without dryer) remained well within or very close to, in the case of $N_2O$, the WMO recommended compatibility goals for independent measurement programs, proving that the test condition induced no significant bias on the sample measurements.

## 1 Introduction

$CO_2$ is one of the most important and extensively monitored greenhouse gases in the atmosphere. Atmospheric $CO_2$ mole fraction measurements provide information that helps understand the continuously increasing mole fractions in the atmosphere due to the combination of human activities, and exchange with the terrestrial- and oceanic components of the global carbon cycle. Further, measurements of the isotopic composition of the atmospheric $CO_2$ provides information about the sources and sinks. $CO_2$ mole fraction can either be continuously measured using instruments capable of performing

continuous-flow measurements in whole air samples e.g., using nondispersive infrared (NDIR) based sensors (Stephens et al., 2011), cavity ring-down spectrometers (Chen et al., 2010) or quasi-continuously by using gas chromatography (van der Laan et al., 2009). Alternatively, discrete air samples can be collected in flasks for later analysis in a laboratory. Flasks are typically filled with ambient air either by opening the valve of a pre-evacuated flask, or by pressurizing a flask with the help of a pump. Under humid conditions, flask sampling requires drying of the sample air to prevent condensation inside the flask

which can affect the $CO_2$ mole fractions as well as for the oxygen stable isotopes composition (Gemery et al., 1996; Trolier et al., 1996).

Since 2009, a substantial effort is undertaken to establish a long-term atmospheric mole fraction $CO_2$ record over the Amazon rain forest. Air samples are collected onboard a small aircraft along a vertical profile from 4.4 km down to 300 m amsl (above mean sea level) at a bi-monthly rate at 4 different sites (Alta Floresta (ALF), Rio Branco (RBA), Santarém

(SAN), and Tefé (TEF)) over the Amazon forest. Additionally, samples are also collected once every month at Manaus (MAN) and over a big flooded area in a different ecosystem at Pantanal (PAN, Mato Grosso state). This unique $CO_2$ program resulted in a number of new insights on *net* carbon exchange from this region (Alden et al., 2016; Gatti et al., 2014; van der Laan-Luijkx et al., 2015) and the measurements continue still. The project "Airborne Stable Isotopes of Carbon from





the Amazon" (ASICA, http://www.asica.eu) aims to add a record of high-precision measurements of the singly-substituted
isotopologues of $CO_2$ to determine $\delta^{13}C$, $\delta^{17}O$, and $\delta^{18}O$ of the atmospheric $CO_2$. Whereas the former isotope potentially
constrains the water-use efficiency of $CO_2$ exchange (Keeling et al., 2017; Peters et al., 2018), the latter two isotope ratios
can be combined into so-called "excess-$^{17}O$ in $CO_2$ ($\Delta^{17}O$)", which has shown early potential to serve as a tracer of leaf-
atmosphere exchange of $CO_2$ (Hoag et al., 2005; Hofmann et al., 2017; Koren et al., 2019), with potential new insights on
the *gross* carbon uptake through photosynthesis of this vast forest area. Removal of water vapour from the sampled air is of
utmost importance though, as oxygen isotopes can easily equilibrate between $CO_2$ and liquid water, destroying the signature
of $\Delta^{17}O$ we are after in ASICA. To perform reliable measurements of $CO_2$ and its isotopologues, an efficient inline dryer
thus had to be included in the sampling system.

Here we describe the development and testing of a field-deployable, consumable-free, power-free, and portable drying
system based on multi-tube Nafion™ gas sample driers (Perma Pure, Lakewood, USA). Nafion is a copolymer of perfluoro-
3,6-dioxa-4-methyl-7-octene-sulfonic acid and tetrafluoroethylene which is known for its unique selectivity and high
permeability for water (Perma-Pure-LLC, 2019). In the form of a membrane, this property enables it to transfers moisture
from the sample gas stream to a counter-flowing dry purge gas stream. The design of the Nafion air dryer, hereon referred to
as NAD, consists of 2 Perma Pure PD-Series gas dryers containing 200 Nafion tubes each in a stainless steel tube shell (PD-
200T-24MSS) placed in series (as shown in Figure 1). A counter flow of dry purge gas within the shell removes moisture
from the sample air stream permeating through the tubing. The dry purge air is provided by feeding the exhaust flow of the
PFP through a 350 g dry molecular sieve (type 3A) cartridge, which effectively removes water to a dew point better than -5
°C at STP conditions. The NAD is designed to be part of the standard sampling system used in Brazil, as well as in the
cooperative air sampling program in the United States, which consists of a "Programmable Compressor Package" (PCP) and
a "Programmable Flask Package" (PFP, Version 3) containing twelve 700 ml (Sweeney et al., 2015) flasks. Since Feb-2018,
the first NAD used also in the tests described here is in actual use at RBA, while a second one (together with a third) will be
deployed at ALF in the second half of 2019.

We present the results of extensive experiments testing the drying capacity, as well as the stability and potential interference
of the NAD on $CO_2$ mole fractions and its singly-substituted isotopologues and on other important atmospheric trace gases
such as $CH_4$, $CO$, $SF_6$ and $N_2O$. The drying capacity and endurance of the Nafion tubes in combination with the molecular
sieve drier cartridge was examined under laboratory test conditions with sample air containing ~4 % $H_2O$ at 33 °C
resembling extreme tropical conditions. Isotopic measurements of sample air fed through the NAD were performed using the
dual laser Tunable Infrared Laser Direct Absorption Spectrometer (TILDAS, Aerodyne Research, Inc.) at Groningen,
specially designed for the measurements of all the three singly-substituted $CO_2$ isotopologues, $^{16}O^{13}C^{16}O$, $^{16}O^{12}C^{17}O$ and
$^{16}O^{12}C^{18}O$ in whole air samples (McManus et al., 2015; Sakai et al., 2017). Finally, we present the results of a series of
performance tests of the NAD in combination with a PCP and PFP sampling system simulating field conditions.



## 2 Experimental methods

### 2.1 Analytical techniques

All continuous observations of $CO_2$ mole fractions in air flowing through the NAD at CIO and INSTAAR were conducted using a cavity ring-down spectrometer for $CO_2$, $CH_4$ and $H_2O$ (CRDS, Picarro, Inc., CA, model G2301) (Crosson, 2008).

The overall measurement precision of the CRDS-systems used was typically <0.02 μmol mol$^{-1}$ (ppm) for $CO_2$, <0.2 nmol mol$^{-1}$ (ppb) for $CH_4$, and <2 nmol mol$^{-1}$ (ppb) for CO.

At the CIO-RUG we employed a dual-laser Tunable Infrared Laser Direct Absorption Spectrometer (TILDAS, Aerodyne Research, Inc.) (McManus et al., 2015; Sakai et al., 2017), which we refer to as the TILDAS-SICAS (SICAS for Stable Isotopes of $CO_2$ measurements in Atmospheric Samples), for the measurements of $^{16}O^{13}C^{16}O$, $^{16}O^{12}C^{17}O$ and $^{16}O^{12}C^{18}O$ in

whole air samples. More details about the TILDAS-SICAS set-up and performance will be described elsewhere in a forthcoming paper. The isotopic composition of the $CO_2$ in the sample gas with respect to the reference gas is determined using the following equations for $^{16}O^{13}C^{16}O$, $^{16}O^{12}C^{17}O$ and $^{16}O^{12}C^{18}O$:

$$\delta^{13}C = \left\{ \frac{\left(\frac{^{16}O^{13}C^{16}O}{^{16}O^{12}C^{16}O}\right)_{Sample}}{\left(\frac{^{16}O^{13}C^{16}O}{^{16}O^{12}C^{16}O}\right)_{Reference}} - 1 \right\} \tag{1}$$

$$\delta^{17}O = \left\{ \frac{\left(\frac{^{16}O^{12}C^{17}O}{^{16}O^{12}C^{16}O}\right)_{Sample}}{\left(\frac{^{16}O^{12}C^{17}O}{^{16}O^{12}C^{16}O}\right)_{Reference}} - 1 \right\} \tag{2}$$

$$\delta^{18}O = \left\{ \frac{\left(\frac{^{16}O^{12}C^{18}O}{^{16}O^{12}C^{16}O}\right)_{Sample}}{\left(\frac{^{16}O^{12}C^{18}O}{^{16}O^{12}C^{16}O}\right)_{Reference}} - 1 \right\} \tag{3}$$


The flask samples from the NAD testing replicating field conditions experiments at INSTAAR were analysed at NOAA-ESRL for $CO_2$, $CH_4$, CO and $SF_6$ on the "Measurement of Atmospheric Gases that Influence Climate Change" (MAGICC) system. $CO_2$ is measured on a LI-COR® non-dispersive infrared analyser with a precision of ±0.03 ppm (Conway et al., 1994), $CH_4$ and CO are measured by gas chromatography followed by flame-ionization detection for $CH_4$ with a precision of

±1.2 ppb (Dlugokencky et al., 1994). CO is measured by Vacuum Ultraviolet Resonance Fluorescence spectroscopy with a precision of ±0.3 ppb (Gerbig et al., 1999). $N_2O$ and $SF_6$ are measured by gas chromatography followed by electron capture detection with a precision of 0.3 ppb and 0.04 ppt, respectively. All gases are measured relative to suites of working standards that are directly linked to the World Meteorological Organization primary standard scales. A copy of the MAGICC system as well as an Aerodyne TILDAS is present at the Greenhouse Gases laboratory: LaGEE at INPE, Sao Jose dos



Campos, Brazil (Gatti et al., 2014) for the analysis of the flask samples collected over the Brazilian Amazon as part of the ASICA-project.

For the analysis of the isotopic composition of $CO_2$ ($\delta^{13}C$ and $\delta^{18}O$) in the PFP samples from the NAD testing experiments at INSTAAR, $CO_2$ was extracted from ~450 ml of sample air. This extracted $CO_2$ was then analysed on a dual-inlet isotope ratio mass spectrometer (DI-IRMS) (Isoprime, Elementar, Middlewich, U.K.) at INSTAAR and compared against working
references linked to the INSTAAR realization of the VPDB-$CO_2$ scale. The precision of $\delta^{13}C$ and $\delta^{18}O$ values of $CO_2$ on the DI-IRMS was ±0.02‰ and ±0.04‰, respectively (Vaughn et al., 2004).

**2.2 Design and operation of the Nafion air drying system**

The NAD was developed and tested at the Centre for Isotope Research (CIO), University of Groningen, the Netherlands. Additional experiments were done at the National Oceanic and Atmospheric Administration Earth System Research
Laboratory (NOAA-ESRL), and the Institute of Arctic and Alpine Research (INSTAAR), University of Colorado, both in Boulder, Colorado, USA. The NAD has been designed to operate together with the NOAA-ESRL air sampling system consisting of a "Programmable Compressor Package" (PCP) for flushing and pressurizing the samples (typically to 275 kPa) and a "Programmable Flask Package" (PFP) containing twelve 700 ml borosilicate glass sample flasks to contain the samples. The top panel of Figure 1 shows a picture of a PFP with the inlet/outlet manifolds and the flasks. All the sample
flasks are connected through the manifold and each flask has its own inlet and outlet valve, controlled by a computer. There are several version of the PFP/PCP available and in use at INPE, some with an outlet manifold (version 2, similar to the one shown in Figure 1) and some without an outlet manifold (version 3) that vented the outlet flow into the PFP box itself. To make the version 3 PFPs compatible with the NAD, an outlet manifold was added to this system which is the one shown in Figure 1.

The NAD contains two Perma Pure PD-Series™ Nafion dryers (PD-200T-24-MSS), a molecular sieve cartridge (type 3A, ~2 mm beads, 350 g), a 2 micron in-line filter (Swagelok, SS-4FW-2), stainless steel tubing and various Swagelok connectors. The middle panel of Figure 1 shows a schematic of all the components inside the dryer along with the flow path. The bottom panel shows a picture of the complete Nafion air dryer system. Sample air enters the system through a tubing connected on Quick-Connect 1 and makes its way through the two Nafion dryers connected in series, into tubing connecting
Quick-Connect 2 to the PCP. The PCP then pushes the air through the PFP. The exhaust of the PFP is then directed back to the NAD through a tube to Quick-Connect 3 where the molecular sieve cartridge is connected. The air then travels through the molecular sieve cartridge in order to dry the exiting air stream going into the purge inlet of the Nafion dryers in a direction opposite to that of the sample flow, and finally it exits the system through the exhaust line connected on Quick-Connect 4. During an actual flask sampling, the exhaust of the flask, which is used as a purge flow, is stopped because the
outlet of the flask is closed to pressurize the flask. We note that the required capture of the outflow on Quick- Connect 3



could be easily accommodated by retrofitting the back-manifold on the PFP (version 3), while the older PFPs (version 2) could not readily be used, partially because of their much higher flow rates of up to 40 L/min. The NAD system is housed inside a rugged case (custom built by Dutycases, Drachten, the Netherlands), as shown in the bottom panel of Figure 1.

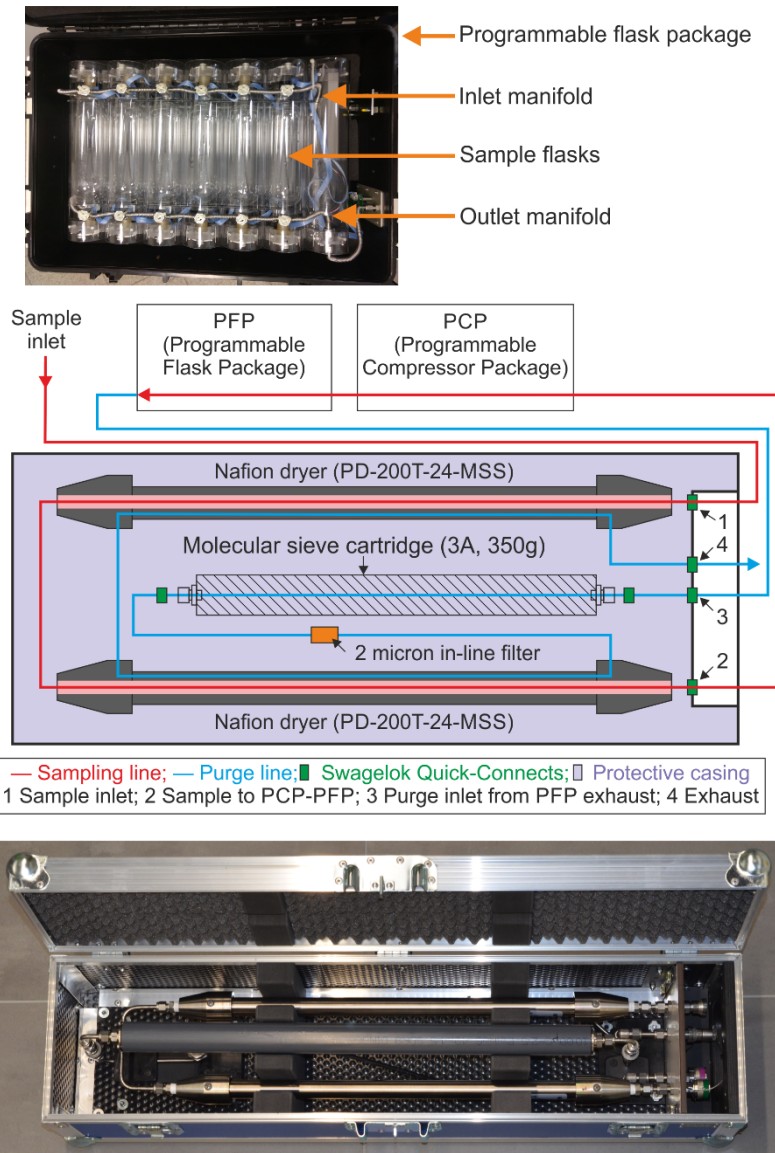

**Figure 1:** *Top: A Programmable flask package showing the flasks and the inlet/outlet manifolds. Middle: A schematic showing the Nafion dryer system, which is housed inside a custom built box, and the sampling order. The Nafion dryer system comprises of two Nafion dryers (Perma Pure PD-200T-24-MSS), a molecular sieve cartridge, a 2 micron in-line filter, and various Swagelok connectors and tubing. Ambient air first passes through the two preconditioned Nafion dryers connected in series (red line), drying the incoming air. It then goes through the PCP which pushes the sample through the PFP. The exhaust of the PFP is connected to the molecular sieve cartridge in the NAD which removes any residual moisture*



*in the air directed into the Nafion purge (blue line). The purge flow through the Nafion is in a direction opposite to the flow of the sample air, and the dry purge air ensures that the Nafion is able to dry at least 230 L of sample air that flows through the system during a typical flight sampling. Bottom: Picture of the Nafion air dryer in a custom built casing (90.5×23.5×22 cm).*

The Nafion dryers were conditioned before every use by passing dry air (from a compressor with a water content typically <0.02 % $H_2O$ by volume) through both the sampling line and the purge line for a period of about 12 hours (typically overnight) at 2 slpm (standard litres per minute). A hygrometer (Rotronic HygroPalm HP22-A) and a CRDS (Picarro, model G2301) were used to monitor the relative humidity of air exiting the Nafion dryers during the conditioning procedure. The drying process was completed when the relative humidity of the outgoing air was <0.02 % $H_2O$ by volume as measured on
the CRDS and reading 0% RH on the hygrometer. The time required for drying the Nafion dryers and the ultimate dryness varied depending on the humidity conditions it was previously exposed to and the moisture content of the drying air used for conditioning the NAD. For atmospheric conditions of ~20 °C and ~2 % $H_2O$, we found that a 10 hour flushing period was more than sufficient to dry the NAD to a $H_2O$ level <0.02 % in the outgoing air. Similarly, the molecular sieve granules were also regenerated after every use by drying them overnight in an open beaker at 100°C. This amount of molecular sieve (~350
g) and the method of regeneration was more than sufficient for experiments that would mimic the sample collection procedure over the Brazilian Amazon, i.e. the molecular sieve was sufficient to remove equivalent quantities of water amounting to ~10 g of $H_2O$. Since the process of adsorption of water on molecular sieve is exothermic, it was conveniently followed using an infrared camera (FLIR i7) to determine an approximate minimum quantity of the adsorbent material required (~175 g).

Following the drying step the NAD system was filled with dry air at ambient pressure. Since all the Swagelok® quick-connect connections used in this system are normally-closed type, the system can thus be stored for several weeks without having to recondition them before use. We successfully tested storage times up to one month in the laboratory. This is especially useful for the sampling strategy employed for the ASICA samples, where the conditioning of the NAD is performed in the laboratory (CCST/INPE, Sao Jose dos Campos, Sao Paulo) and is then stored for a few days before it is
shipped to a site for sample collection. The NAD and the PFP are sent to the sample site together for sample collection and then both are returned to the laboratory for analysis and reconditioning. Note that the PCPs remain at the airport, and residual water vapour in this component (as well as in the tubing of the aircraft) needs to be removed during the system start-up tests on the ground, performed prior to each flight after a preconditioned NAD is attached. During this pre-flight test, ~5 L of outside air is flushed through the full inlet system (all lines + PFP manifold + PCP + NAD) while the humid outside air is
pre-dried with a small hand-held mol-sieve 3A cartridge attached to the wing inlet, to limit loss of NAD-capacity before take-off and actual flask sampling.



## 3. Results

### 3.1 Development and design of the Nafion Air Dryer

When atmospheric samples are collected under humid conditions, there is a risk that the water vapour in the sample leads to condensation on the walls of the inlet tubing or flask especially at higher than ambient sampling pressures. This leads to an oxygen isotopic equilibration between $CO_2$ and $H_2O$, effectively setting the $\delta^{18}O$ and $\delta^{17}O$ in $CO_2$ to that of the more abundant water vapour. To avoid condensation of water inside the PFP sampler, it is necessary to dry the samples to a sufficiently low dew point before storage. For the ASICA program air samples are collected over the Brazilian Amazon region where the air can be close to saturation (>90% relative humidity) at temperature up to 35 °C. These samples are susceptible to condensation when brought back to the laboratory with a typical indoor temperature of ~20 °C.

For the ASICA project, typically 12 flasks are filled with dried air. Flask filling is initiated by toggling a switch that initiates the pumps in the PCP and switches flask valves in the PFP. The inlet tubing, NAD system, PCP and PFP manifold is first flushed with 5 L of air, followed by opening of the valves and flushing of the flask with 10 L of air. A sample is collected by closing the down-stream flask valve and pressurizing the flask to 275 kPa before closing the upstream valve (corresponding to ~1.9 L of air at STP). The total amount of air that is dried per sample, from flushing to sampling, is around ~17 L, adding up to ~204 L for one flight of 12 samples, excluding the pre-flight test which is done with pre-dried air.

The first drying method we tested was an in-line magnesium perchlorate ($Mg(ClO_4)_2$) packed cartridge to dry humidified air, which at first seemed to be an easier alternative to the NAD to remove water vapour from the sample stream onboard a small aircraft. Magnesium perchlorate is a desiccant, capable of drying air samples without affecting its composition (notably of $CO_2$ and its isotopic composition). It can be regenerated by heating at 220°C ($Mg(ClO_4)_2.6H_2O \rightarrow Mg(ClO_4)_2$) under vacuum. Theoretically, to dry around 200 L of humidified air containing 4% water vapour, a minimum of 12.4 g anhydrous $Mg(ClO_4)_2$ would be sufficient. However, the use of magnesium perchlorate, includes a number of disadvantages that led us to look for an alternative better suited for the ASICA program. The first disadvantage of using magnesium perchlorate concerns safety and health hazards inherent in perchlorates. Perchlorates are stable at normal temperatures, but when they are exposed to high temperatures e.g. in case of a fire, they thermally decompose and become explosive. Secondly, in case of an accident, exposure to perchlorates can cause serious skin, eye and respiratory irritations. Hence, usage onboard an aircraft is mostly prohibited or at least restricted to an amount too small for our purpose. Another drawback of using magnesium perchlorate is that it tends to change into a thick slurry when retaining significant amounts of water which eventually restricts the sample flow and in the worst case could block the flow completely. Finally, it is difficult to regenerate it to its original grain size after usage, so typically each flight would require a fresh batch.

As a result we decided to move away from $Mg(ClO_4)_2$ to dry our sample air and started experimenting with multi-tube Nafion gas sample driers from Perma Pure. Due to the relatively high flow rate of the PCP-PFP sampling system of up to 15



L/min we choose to use the 24-inch Perma Pure PD-Series gas dryers containing 200 Nafion tubes each in a stainless steel
tube shell designed for high flows up 40 L/min. To dry a sample flow a counter flowing dry purge air is needed. According

to manufacturer's recommendations, optimal result with one PD-Series tube would be achieved when purge air of -40 °C
dew point can be offered at a flow rate of two to three times the sample flow at a pressure equal or lower than the sample
flow (Perma-Pure-LLC, 2019). However, this would require an additional dry air tank containing at least 600 L of
compressed air on board each flight which is undesirable both from a logistic and safety point of view. When no dry purge
gas is available, one can choose to reuse the sample gas itself after it is partially dried passing through the Nafion tube (Welp

et al., 2013), or for a lower dew point, with an additional water trap such as freeze dryer/molecular sieve to remove the
remaining water before it is reused as purge gas (e.g. (Neubert et al., 2004; Stephens et al., 2011)). We choose to use
molecular sieve (type 3A) as a drying agent because it is non-toxic, economical, and reusable.

Besides the number of Nafion tubes inside the shell and the dew point of the purge air the performance of the Nafion tube is
dependent on the dryer length and both the sample and purge flow rates. In our set-up, the sample flow rates are typically

~12 L/min (sample flow = purge flow). According to the PD-200T-24MSS specifications of the manufacture (Perma Pure,
USA), a sample stream of 12 L/min with a dew point of 20 °C would require a dry purge flow at two times the sample flow
rate to get to a dew point of −12 °C. For the samples collected over the Amazon basin, the dew point could be as high as 30
°C and the maximum possible flow rate we can offer to the purge line is equal to the sample flow rate. Additionally, since
the samples are compressed to 275 kPa, the water vapour content in the samples have to be even lower than at STP

conditions as discussed earlier. Thus a single 24" Nafion dryer wouldn't be sufficient to achieve the required lower water
vapour content in all the 12 sample flasks. Hence we decided to instead use two 24" Nafion dryers in series to increase the
effective interaction length to 48", thus ensuring acceptable levels of water vapour content in the sampling flasks. A previous
study (Gemery et al., 1996) recommended that to obtain reliable measurements of oxygen isotopes in $CO_2$, one needs to dry
flask samples to better than 2 °C dew point for flasks filled to atmospheric pressure (RH ≈ 30 % at 20 °C). The authors noted

that the largest deviations in $\delta^{18}O$ were observed only when the relative humidity in the flasks were above 100 % at
conditions where water would condense on the wall of the flask. No significant deviations were observed for flasks
containing lower than 60 % relative humidity, whereas small deviations were observed between 60-100 % RH. The
deviations observed above 60 % relative humidity gradually increased and were >0.5 ‰ above 80 % relative humidity.

We used this study to set our boundary conditions in order to achieve reliable $\delta^{18}O$ measurements. Thus, a flask filled at 275

kPa with 100 % RH (at 20 °C) corresponds to a dew point temperature of ~5 °C; and with 60% RH (at 20 °C) to a dew point
temperature of ~−2 °C. We have been monitoring the temperature experienced inside the PFP, since the time it is sent to the
sample collection site to the time it is brought back to the lab, for a period of approximately two years, with the help of a
portable temperature sensor (Omega engineering, OM-EL-USB-1). This temperature profile data indicates that the minimum
temperature the PFP experience in the Amazonian sample collection sites is about 11 °C. As discussed and shown later in



this manuscript, most samples collected in the Amazonia have a water vapour content lower than −2 °C dew point (~60 % RH) and none would exceed the 5 °C dew point (~100 % RH) limit at 275 kPa flask pressure. In summary, for an optimal performance under tropical saturated conditions we decided to use the PD-200T with a total length of 48” (~140 cm). For practical reasons we chose to use two of the 24” (~70 cm) long PD-200T-24MSS, which can be placed in series (in a parallel configuration) to save space.

The amount of molecular sieve needed to sufficiently dry the purge gas was initially set on ~150 g. From our experiments testing the drying capacity of the double Nafion tube we found that a minimum of 175 g of dry molecular sieve was needed for drying a minimum of 200 L of air at moderate conditions of ~22 °C and 70 % RH down to <1 % RH. For a safe margin with respect to the humid tropical conditions we meet over the Amazon region we decided to double the amount of molecular sieve to 350 g. Our work on testing the drying capacity of the NAD is further elaborated in the next section.

**3.2 Drying capacity of the NAD**

The drying capacity of Nafion is dependent on the dryness of the purge flow and the individual Nafion tubes. The hydrophilic properties of sulfonated tetrafluoroethylene polymer, of which Nafion is formed, absorb water molecules at the humid side and releases it at the dry side creating a permeation of water from the wet-side to the dry-side. When the Nafion material is not properly dried before usage, the material will remain saturated with water molecules for a longer period of

time and will thus limit its total capability to dry the sample air stream. Hence, before each experiment or operation with the NAD it was essential to dry the Nafion tubes by flushing them with dry compressed air at a rate of 2 L/min at STP for about 12 hours. After this time, the outflow of the NAD would show no remaining water vapour, i.e. the outflow equals the inflow water vapour content indicating the Nafion is dry.

Here we describe the results of a specific experiment where we mimicked the sample collection process during a flight under

tropical conditions to demonstrate the usability and the performance of the NAD. This experiment also allowed us to determine the total amount of water the NAD was capable of removing in its current set-up. The experiments were performed in a greenhouse test facility at the biology department of the University of Groningen where the water content of the sampled air could be set to 3.7-3.9 % at ~33 °C mean temperature. As described earlier in section 3.1, a sampling sequence using the PCP-PFP system consist of 5 L of air for flushing the PCP-PFP inlet and manifold, followed by flushing

the sample flask with 10 L of air, and ending with pressurizing the sample flask up to 275 kPa before closing the upstream valve, corresponding to ~1.9 L of air at STP in the flask. The complete filling of a flask was simulated by flushing the NAD periodically at 6 L/min with ambient air from the greenhouse for a duration of 3 minutes, depicted in Figure 2. The bottom left axis in Figure 2 shows the water vapour content, as measured by the Picarro CRDS; the top left axis shows the corresponding dew point temperature at 275 kPa; and the right axis shows the total volume of air processed through the

NAD. The first 50 seconds correspond to the inlet and manifold flushing (blue points in the dew point data), the next 100



seconds refer to the flushing of the flask + inlet and manifold (red points), and the filling resembled the last 20 seconds (green points). Although we tried to process approximately 17 L of humidified air in every step, similar to a real sampling scenario, there was a slight excess air that was processed in every step (pink points). With a targeted maximum dew point of −2 °C, we found that the drying unit is capable of drying 4 samples, corresponding to approximately ~85 L of air containing

3.7-3.9 % $H_2O$. The drying capacity of the NAD is related to both the rate of saturation of the Nafion tubes and also the amount of molecular sieve used. At high humidity conditions both the Nafion tubes and the molecular sieve cartridge gets saturated very fast and ultimately affects the number of samples that can be processed. We note that this can be considered as a worst case scenario relative to the average conditions met at the ASICA sampling program where typically 6 out of 12 flasks are filled below 2 km altitude in the tropical boundary layer and the remaining flasks are collected in the dryer middle

troposphere up to about 5 km altitude.

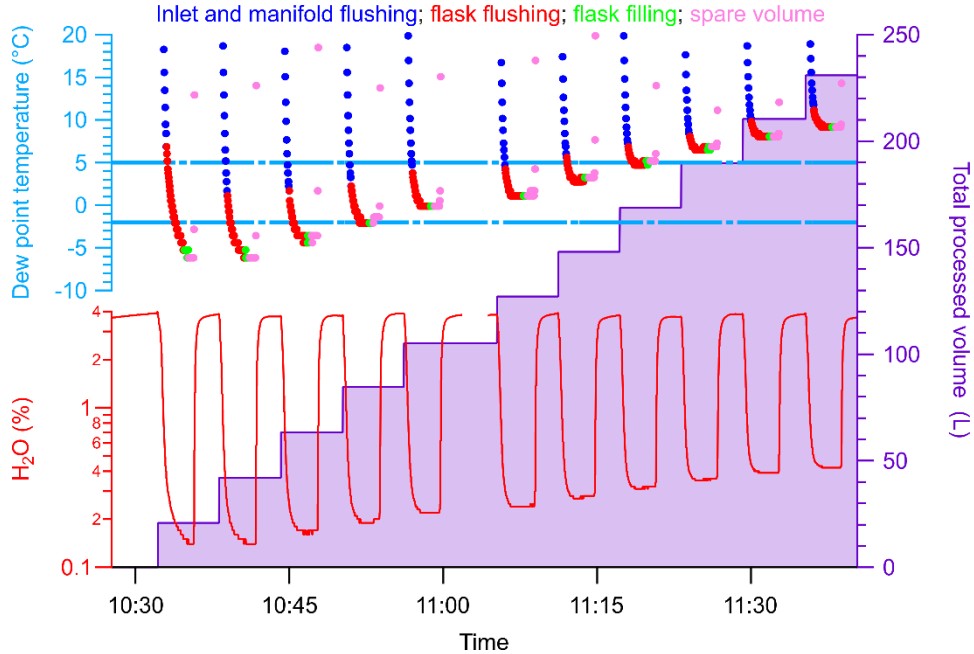

***Figure 2****. A time-series mimicking the sample collection process during a flight to demonstrate the usability and the response of the Nafion dryer under conditions similar to that of the tropics (sample air with 3.7-3.9 % water vapour content at ~33 °C). The bottom panel (left) shows the water vapour content in the sampled air, the top panel (left) shows the*

*corresponding dew point temperature (at 275 kPa) and the panel on the right shows the total processes volume. Periodically, humid air from the test facility was passed through the NAD, indicated by the dips in the water vapour content (%) of the sampled air, to simulate a complete filling step of a PFP flask. These steps are indicated in the dew point plot as follows: 1) flushing of the inlet and the manifold volume (blue points); 2) flushing of the inlet and manifold volume + the flask (red points); 3) filling of a flask (green points). A slight excess air was processed in every passing through the Nafion*

*dryer, indicated by the pink points. The targeted dew point temperature was −2 °C and as can be seen, the drying unit is capable of drying approximately 85 L of air containing 3.7-3.9 % $H_2O$.*

From the data presented in Figure 2, we estimated the total amount of accumulated water after which the NAD is not capable

of delivering samples dryer than the −2 °C dew point (at 275 kPa) is 2.4 – 2.5 g. In Figure 3 we present a typical sampling

profile over the Amazon as a function of altitude and the estimated amount of water removed by the NAD system. The

bottom axis shows the amount of water removed (g) by the dryer and the top axis shows the water vapour content in the

sample at a given altitude (blue solid line). The water vapour content at each 1 km altitude bin were taken from sounding

data (Soundings-Data, 2019) collected during the wet seasons over Rio Branco and Santarém and were the maximum values

observed in that altitude bin. Of course, the water vapour content gradually decreases as a function of altitude, and the

calculated amount of water removed, as shown in Figure 3, is thus an overestimate.

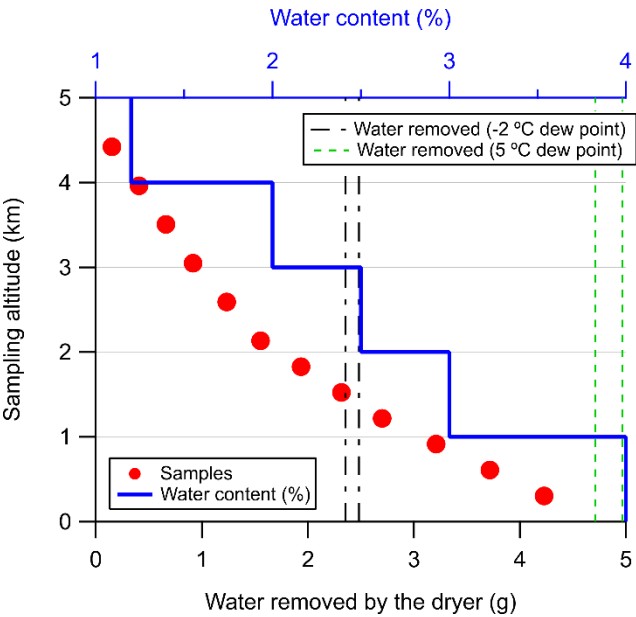

**Figure 3**. *A simplified estimation of the total amount of water that the Nafion dryer has to remove during a typical flight over the Amazon. Twelve samples are collected gradually from a height of about 4.4 km down to about 300 m (amsl). The water vapor content in the air at a given altitude is shown with the solid blue line. This value has been assumed constant over every 1 km altitude bin, and was the maximum value observed in the sounding data, collected during the wet season*
*over Rio Branco and Santarem, over the altitude bin. The black and green dashed lines indicate the lower and upper boundaries for the number of samples that can be collected with a dew point lower than −2 °C (at 275 kPa) and 5 °C (at 275 kPa), respectively. These lower and upper boundaries were estimated from the experiment shown in Figure 2.*

Sampling is gradually carried out from an altitude of about 4.4 km down to about 300 m and 12 samples are typically

collected during such a flight and are shown with the red points. The two black and green dashed lines indicate the lower and

upper boundaries, determined from the data presented in Figure 2, for the number of samples that can be collected with a

dew point lower than −2 °C (60 % RH) and 5 °C (100 % RH), respectively. This estimation shows that for a typical flight,

the accumulated water in the NAD stays well below the worst case scenario approximated by the greenhouse experiments

where only 4 samples could be collected before reaching the −2 °C dew point threshold. Figure 3 further indicates that at





least 8 samples can be collected under typical flight conditions with dew points below −2 °C and the last four with dew

points well below 5 °C.

### 3.3 Effect of the NAD on CO₂ isotopic composition

To determine the effect of the Nafion dryer on the isotopic composition of $CO_2$ in an air sample, a "Zero-Enrichment" (ZE) experiment was performed, similar to the ones performed with a dual-inlet IRMS (Wright et al., 1983). For the ZE experiments, air from a compressed air tank was treated both as a reference gas (unprocessed: dry) as well as a sample gas

(processed: humidified). Thus, in a ZE experiment, if the isotopic composition of the reference gas is identical to that of the isotopic composition of the sample gas, the resultant difference is zero ($^{17}\delta$, $^{18}\delta$, and $^{13}\delta = 0$ ‰, see equations 1-3 section 2.1), indicating no effect.

To determine the isotopic composition of $CO_2$-in-air and the influence of the Nafion dryer on the stable isotopes of $CO_2$, we used our TILDAS-SICAS which was designed to detect the $CO_2$ isotopologues $^{16}O^{12}C^{16}O$, $^{16}O^{13}C^{16}O$, $^{18}O^{12}C^{16}O$, and

$^{17}O^{12}C^{16}O$ in whole air samples (described in more detail earlier in section 2.1 Analytical techniques). All the measurements performed on the TILDAS-SICAS were static measurements, i.e., a specific volume of air is introduced into the optical cell and a measurement is then performed for a period of 60 s after which the optical cell is evacuated and the next sample is introduced. This measurement scheme allowed a semi-continuous measurement mode for all the stable isotopes of $CO_2$ to investigate the effect of the NAD on the isotopic composition of the downstream $CO_2$ as a function of time. For the reference

measurements, the gas was directly fed into the TILDAS-SICAS for the measurements of the mole-fractions of all the stable isotopologues of $CO_2$. For the sample measurements, air from the same cylinder was either sent into the TILDAS-SICAS through the NAD (dry mode) or it was first humidified (~2 %) and then dried using the Nafion dryer (wet mode), before sending the air stream into the TILDAS-SICAS for the measurements of the mole fractions of all the previously mentioned isotopologues of $CO_2$. The sample air stream was humidified at room temperature, by passing the dry cylinder air through a

bubbler containing demineralized water. The nozzle of the bubbler was constructed using a sintered glass filter to maximize the water content in the downstream sample air, reaching a maximum of ~2 % at room temperature.

One such experiment is illustrated in Figure 4, where the $CO_2$ mole fraction is shown in the top panel, and the calculated $\delta^{17}O$, $\delta^{18}O$, and $\delta^{13}C$ are shown in the subsequent panels below the $CO_2$ mole fraction panel. The calculated $\delta^{17}O$, $\delta^{18}O$, and $\delta^{13}C$ values for the sample measurements are determined with respect to the reference measurements performed before and

after each sample measurement. This measurement strategy is similar to the ones used for IRMS based measurements which eliminates systematic instrument drifts during a reference-sample-reference measurement set. The first part of the experiment (shown in red background) was performed in dry mode, and the second part (shown in blue background) was performed in wet mode. The section between the dry and the wet mode, shown with a white background, denotes a stabilization period during which the cylinder air was humidified and the wet air stream was passed through the NAD. The total volume of air





processed in the dry mode and wet mode (including the stabilization period) was 162 L and 174 L, respectively, at a flow
rate of 4.5 L/min. As can be seen from the data in Figure 4, the isotopic composition of $CO_2$ in the cylinder air is not altered
by the NAD, both in dry and wet conditions. One sample measurement (measurement number 16) in the dry mode was
affected by an unknown cause and was considered an outlier, thus not included in the analysis. The mean of the ZE
measurements (± SE), corresponding to the dry and wet mode, for the three singly-substituted isotopologues of $CO_2$ are also

shown in Figure 4. We note that the flow rate used during this experiment (4.5 L/min) is significantly lower than the flow
rates typically used during sample collection (~12 L/min). This clearly demonstrates that under laboratory test conditions the
NAD has negligible effect on the isotopic composition of $CO_2$, even with significantly longer residence times in the Nafion
tubes.

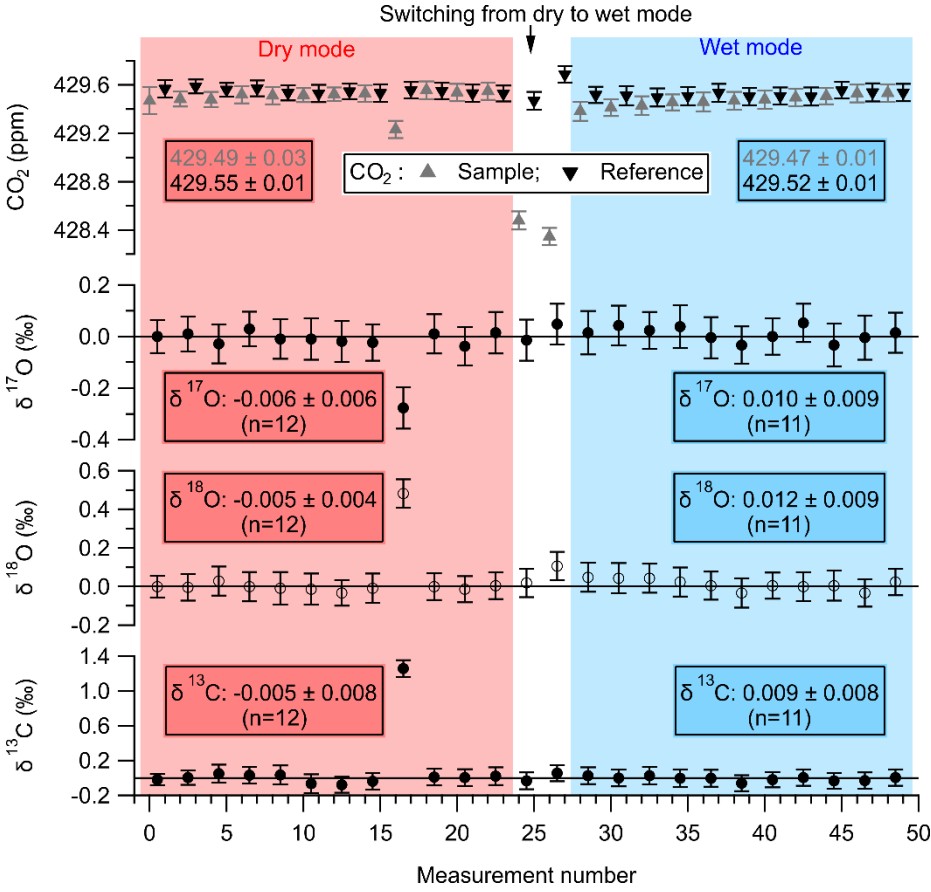

***Figure 4****. A zero-enrichment experiment performed with the TILDAS-SICAS to test the effect of the NAD on the $CO_2$ mole
fraction and the stable isotopes of $CO_2$. The top panel shows the $CO_2$ mole fraction, where the grey triangles represent the
sample measurements and the black inverted triangles represent the reference measurements. The other three panels show
the isotopic composition of the sample gas determined with respect to the reference gas. The reference gas is identical to the
sample gas, the only difference being that the sample gas was measured after it had passed through the NAD. The first part*

*of the experiment represents dry mode (shown in red background) where, for sample measurements, dry air from a cylinder
was passed through a preconditioned NAD and was then introduced into the TILDAS-SICAS for measurements. In the latter*





*half of the experiment in wet mode (shown in blue background), the dry air form the cylinder was first humidified (~2 %*
*$H_2O$) and then dried through the NAD before introducing it into the TILDAS-SICAS for measurements. As can be seen, the*
*NAD does not affect the measurements of $CO_2$ mole fractions and its stable isotope composition.*

**3.4 Effects on other atmospheric trace gases**

The performance of the NAD was tested in conjunction with the NOAA-ESRL PCP-PFP sampling system simulating real
sampling conditions at the NOAA-ESRL and INSTAAR laboratories in Boulder, Colorado. The objective of these
experiments was to test the effect of a known air sample, when collected into the PFP flasks after being dried by the NAD,
on $CO_2$ and its two singly-substituted isotopologues ($^{13}C^{16}O^{16}O$ and $^{18}O^{12}C^{16}O$), as well as on other greenhouse gases
measured on the MAGICC system ($CH_4$, CO, $N_2O$, and $SF_6$). For the known air sample, we used two different calibrated dry
whole air samples in a 29 L Luxfer aluminium cylinder compressed at ~140 bar. The test gas was flushed and compressed by
the PCP into a 12-flask PFP using the set-up illustrated in Figure 5. To provide a homogeneous sample just above
atmospheric pressure (~1.4 kPa) the dry test gas from the cylinder was flushed into a 300 L buffer volume at a rate of 12-13
L/min. This flow rate was a bit higher than the flow rate of the PCP-PFP during flushing and sample collection, allowing a
small excess flow of ~0.5 L/min (measured with a rotameter). A Picarro CRDS was used to measure the stability of $CO_2$,
$CH_4$, and $H_2O$ in the excess flow exiting the buffer volume. To humidify the air we used a bubbler containing demineralized
water at a laboratory temperature of 23 °C, which resulted in a water vapour content of ~1.4 % (corresponding to ~40 %
relative humidity) by volume as indicated by the Picarro CRDS instrument. To check the relative humidity and temperature
of sample air in the buffer volume a hygrometer (Rotronic HygroPalm HP22-A) was also placed in the excess flow line. The
flasks were prepared for sampling by flushing them with dry air, followed by filling them with synthetic air containing ~350
ppm $CO_2$ at atmospheric pressure. Sampling is started by toggling a switch that initiates the pumps in the PCP and opens
valves in the PFP to flush the manifold and flask. After flushing, the flask is pressurized to 275 kPa. We noted that the
inclusion of the NAD imposed more drag on the PCP pumping capacity reducing the flushing flow from ~15 L/min to ~12
L/min. We conducted three successful experiments of filling one 12-flask PFP and used two compressed air cylinders as
sample gas during this experimentation period.



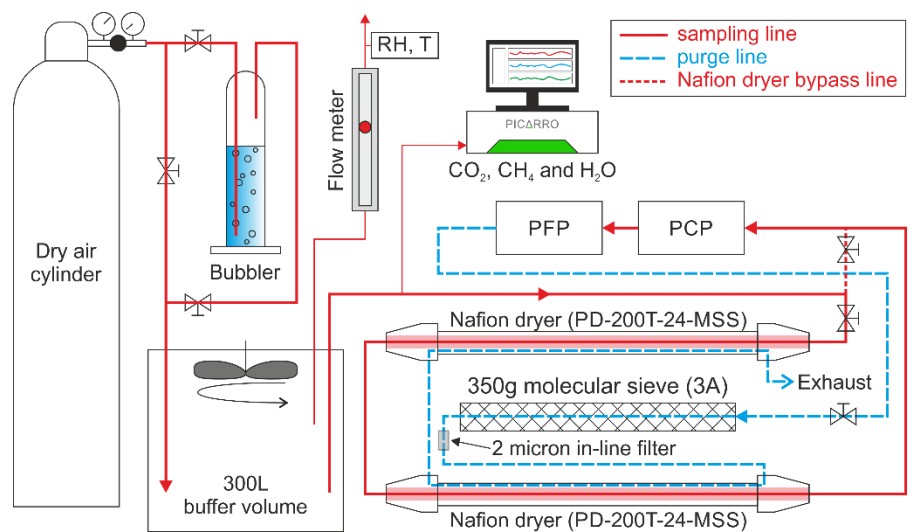

**Figure 5.** *Schematic of the setup used at INSTAAR to determine the effect of the Nafion Air Dryer on the mole fraction determination of CO, CH₄, N₂O and SF₆ along with CO₂ and its singly-substituted isotopologues. Dry sample air from a calibrated cylinder was flushed into a buffer volume either directly or through a bubbler to humidify it up to ~1.4 % water vapour by volume at an average temperature of 23 °C. A NOAA-ESRL PCP-PFP system could be flushed with either dry or humidified cylinder air including or by-passing the NAD. Flow from the cylinder was adjusted such that there was always an overflow of air from the buffer volume, where relative humidity of the sample was constantly monitored. A Picarro analyser was used to monitor the stability of CO₂, CH₄ and H₂O mole fractions in the gas exiting the buffer volume, during all the tested experimental conditions, a summary of which is shown in Figure 6.*

In these experiments, we tested 4 different conditions by filling a set of three flasks under the following conditions: (A) dry air-without dryer, (B) dry air-with dryer, (C) wet air-without dryer, and (D). wet air-with dryer. With respect to applying these compatibility goals it should be mentioned that these precisions should be seen as the scientifically desirable level of compatibility for concurrent measurements of well-mixed background air by different laboratories, while they may not be the currently achievable best 1-σ measurement uncertainty (GAW Report No. 242, 2017). Indeed, a recent study by (Zellweger et al., 2019) indicated that the N₂O network compatibility goal of 0.1 ppb remains quite challenging to meet even with current state-of-the-art measurement techniques. We note that the NAD dried air samples from the ASICA program are analysed on a copy of the MAGICC system and on an Aerodyne TILDAS system (for all the singly-substituted isotopologues of CO₂) similar to the one at CIO, University of Groningen, both located at the Atmospheric Chemistry laboratory of INPE, San Jose dos Campos, Brazil. In Figure 6 we present the results of 5 cases comparing a base condition with a test condition in the following order: Case 1: condition B − condition A, Case 2: condition C − condition A, Case 3: condition D − condition A, Case 4: condition D − condition C, and Case 5: condition D − condition B. We show the mean difference and the corresponding 1-σ standard deviation (error bar) indicating the spread in the results, while the dashed line indicates the WMO network compatibility goals. Table 1 summarizes the tests and the information we get from each case.



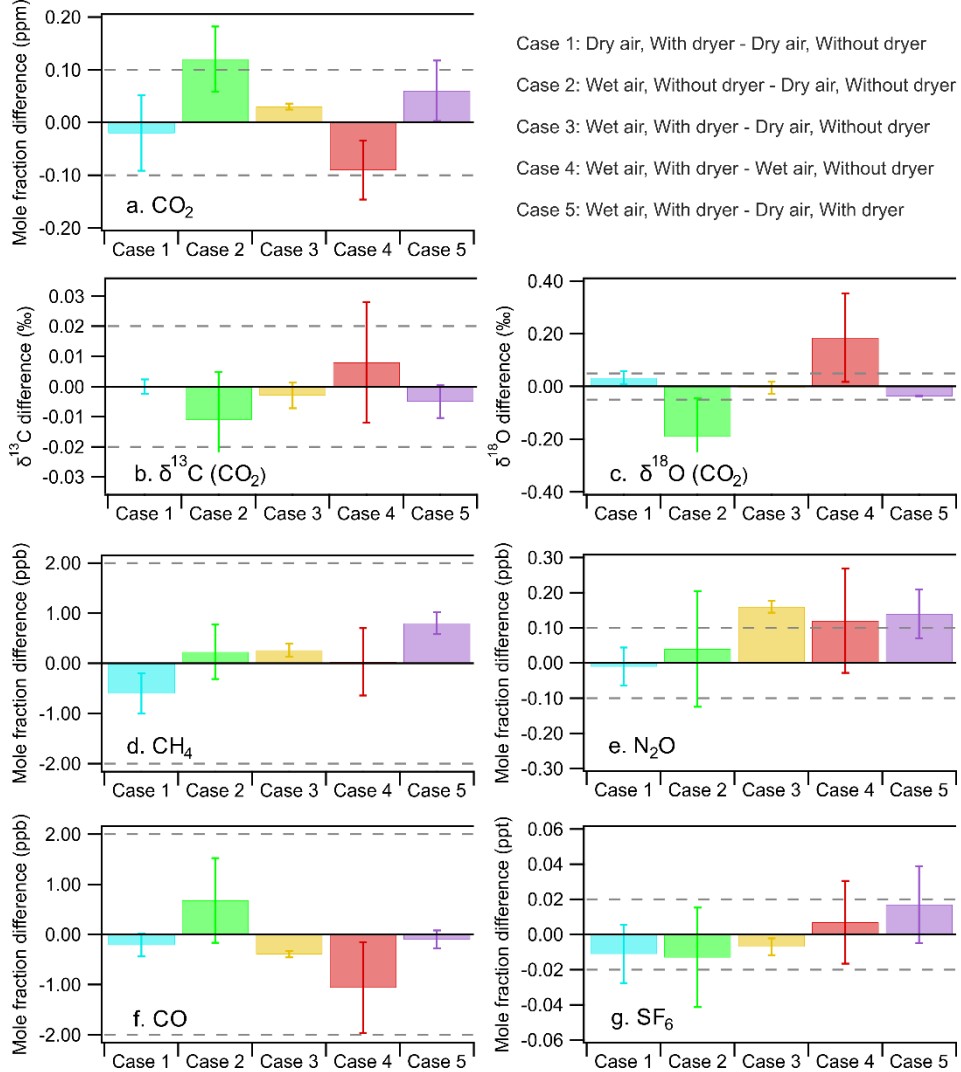

*Figure 6. Overview of results to determine the effect of the NAD on the mole fraction determination of CO, CH$_4$, N$_2$O and SF$_6$ along with CO$_2$ and isotopic composition of its two singly-substituted isotopologues ($^{16}O^{13}C^{16}O$ and $^{16}O^{12}C^{18}O$). The dashed lines denote the WMO network compatibility goals for the different species. We note that the compatibility goal for N$_2$O should be seen as a lower limit target value. We present 5 test cases to illustrate the measurement biases introduced if samples are not sufficiently dried. Case 1 shows the effect of the NAD in dry conditions, whereas cases 3 and 5 show the difference in sampling of wet sample air dried with the NAD before being compressed into the sample flask with respect to dry air sampling. The worst sampling conditions with the largest bias are associated with using humidified sample air (without NAD) shown in cases 2 and 4. When the difference remain within the WMO network compatibility goals, we concluded that the measured difference is not significant (notably for cases 1, 3 and 5 for most species).*

Case 1 ("Dry air-with dryer" – "Dry air-without dryer") should indicate if there is any bias introduced by the Nafion dryer in the sampling line. Indeed, the results show that the differences for "condition B − condition A" are all very small and that they stay well within the WMO network compatibility goals. In cases 2 and 4 we compare wet samples (without dryer)





containing ~1.4 % water vapour with either dry air without the dryer or dry air with the dryer, respectively. As expected we find that $CO_2$ and its isotopologues (notably $\delta^{18}O$) are affected. In cases 3 and 5 we compare the wet air-with dryer (condition D) with dry air-without dryer (condition A) and with dry air-with dryer (condition B), respectively. These cases

prove the benefit of using the NAD by showing that the differences in "condition D − condition A" and "condition D − condition B", notably those of $CO_2$ and its singly substituted isotopologues, are well within the WMO network compatibility goals.

| Species | Case 1 "Dry air, with dryer" – "Dry air, without dryer" | Case 2 "Wet air, without dryer" – "Dry air, without dryer" | Case 3 "Wet air, with dryer" – "Dry air, without dryer" | Case 4 "Wet air, with dryer" – "Wet air, without dryer" | Case 5 "Wet air, with dryer" – "Dry air, with dryer" |
|---|---|---|---|---|---|
| $CO_2$ (ppm) | -0.02 ± 0.07 | 0.12 ± 0.06 | 0.03 ± 0.01 | -0.09 ± 0.06 | 0.06 ± 0.06 |
| $^{16}O^{13}C^{16}O$ (‰) | 0.000 ± 0.002 | -0.011 ± 0.016 | -0.003 ± 0.004 | 0.008 ± 0.020 | -0.005 ± 0.005 |
| $^{16}O^{12}C^{18}O$ (‰) | 0.033 ± 0.025 | -0.190 ± 0.144 | -0.005 ± 0.023 | 0.185 ± 0.167 | -0.037 ± 0.002 |
| $CH_4$ (ppb) | -0.60 ± 0.40 | 0.23 ± 0.54 | 0.26 ± 0.13 | 0.03 ± 0.67 | 0.80 ± 0.22 |
| $N_2O$ (ppb) | -0.01 ± 0.05 | 0.04 ± 0.16 | 0.16 ± 0.02 | 0.12 ± 0.15 | 0.14 ± 0.07 |
| CO (ppb) | -0.21 ± 0.23 | 0.68 ± 0.84 | -0.39 ± 0.06 | -1.06 ± 0.91 | -0.10 ± 0.18 |
| $SF_6$ (ppt) | -0.011 ± 0.016 | -0.013 ± 0.028 | -0.007 ± 0.005 | 0.007 ± 0.023 | 0.017 ± 0.022 |

*Table 1. Overview of results from Case 1 to Case 5 (test result – reference value) to determine the effect of the NAD on the*
*mole fraction determination of $CO_2$, CO, $CH_4$, $N_2O$, CO, $SF_6$, and on the isotopic composition of the two singly-substituted isotopologues $^{16}O^{13}C^{16}O$ and $^{16}O^{12}C^{18}O$ of $CO_2$. The measurement uncertainty denotes the 1-σ standard deviation of typically 3 samples per condition.*

## 4. Discussions and conclusions

Since unbiased measurements of $CO_2$ mole fraction and its isotopic composition in whole air samples demand collection of
very dry sample air, we tested and present here the results of a Nafion based drying system. Nafion dryers are an excellent alternative to chemical and recirculating chiller based dryers for mobile sampling platforms. For example most chemical dryers either alter the chemical composition of the sample air, or are considered hazardous from a safety standpoint, especially when they are used onboard an aircraft. On the other hand, recirculating chiller based dryers are very efficient but are large and extremely energy demanding, which makes their usage on light aircrafts logistically undesirable. Nafion based
drying systems offer a consumable-free, reusable, and a field-deployable alternative, which does not necessitate incorporating hazardous chemicals and also eliminates the use of any power onboard an aircraft. In this work we tested the





NAD which is configured for use with the PCP-PFP system from NOAA-ESRL, although the use of our system is not limited to that sampling platform.

During the development phase we learned that to achieve the best performance from the dryers in the configuration we wanted to use, they had to be conditioned before every use. Conditioning was performed by flushing the Nafion drying tubes and the purge volume overnight with dry air at ~2 L/min, and a successful conditioning was achieved when the water content in the dry air entering the dryer was equal to the water content in the air exiting the dryer. Similarly, the molecular sieve granules were also dried following every use by baking them overnight in an oven set at 100 °C. Since all the end connectors on the NAD are normally-closed Swagelok Quick-Connect connectors, the system is filled with dry air at ambient pressure and stored. We performed a storage stability check over a period of one month and the results indicated that the NAD, if stored in dry conditions, would perform similarly to one freshly conditioned. This property is particularly beneficial for the sampling conditions in Brazil because the conditioning step is performed in the lab few days before the PFP and the NAD are shipped to the sample collection site.

In the ASICA program, the goal is to perform measurements of $CO_2$ mole fraction and high precision measurements of all the singly-substituted isotopologues of $CO_2$ to constrain the gross primary production and its response to droughts for the Amazon basin. In this program, twelve samples are collected per flight at altitudes between 300 m and 4.4 km amsl. To achieve high precision measurements of the isotopic composition of $CO_2$ in the whole air samples, the collected sample air must be dry. Thus the next requirement was to estimate the water removal capacity of the NAD and estimate the number of flasks that could be filled with a targeted dew point of −2 °C (at 275 kPa). From the experiment presented in Figure 2, it is evident that the response of the NAD is almost instantaneous and approximately 85 L of air, containing 3.7-3.9 % $H_2O$, can be processed within the targeted dew point of −2 °C. In Figure 3, we estimated the number of flasks that can be sampled within the targeted dew point temperature during a typical flight above the Amazon. According to these estimates, we can at least collect 8 flasks with water vapour content below −2 °C dew point temperature (at 275 kPa, 60 % RH), whereas the rest of the flasks would still contain water vapour below 5 °C dew point temperature (at 275 kPa, 100 % RH). As discussed earlier, a previous study (Gemery et al., 1996) showed that flask samples containing air with relative humidity below 60% are generally not affected and gradual biases are observed in the 60-100 % RH range. This would thus indicate that the first eight samples collected between 4.4-1.5 km should be free from any bias introduced from the water vapour content in the flask. But the last four samples collected between 1.5-0.3 km could be potentially biased towards the isotopic composition of the water it is exposed to, although this bias should be gradual due to the continuous increase of the water vapour content as more samples are processed through the NAD. This prediction, shown in Figure 3, is of course based on the experiment shown in Figure 2, where the NAD was directly and continuously exposed to air containing ~4 % water vapour. This scenario would saturate the NAD much faster and the ability to dry the Nafion tubes through the purge side would be slower. Whereas, in a real sampling scenario, the NAD is not exposed to high water vapour content as soon as sampling begins and



thus the purge side could still dry the Nafion tubes longer than in the former case. This would then lead to an increase in the
estimated quantity of water removed by the NAD before crossing the 60 % RH limit (−2 °C dew point) in the sampled
flasks, as shown in Figure 3, and would thus encompass more than 8 samples within this limit and would also lower the
biases for the ones outside the limit.

The next requirement was to establish if the NAD was inert for the gases-of-interest and did not alter the isotopic
composition of $CO_2$ while sampling. To understand the effect of the NAD on the isotopic composition of $CO_2$, we performed
a semi-continuous zero-enrichment experiment with the TILDAS-SICAS instrument in our laboratory. In such an
experiment, the same gas is treated both as a reference and a sample gas, where the reference stream is unprocessed and the
sample stream is processed. Thus, a zero-difference between the reference and the sample stream would indicate that the
processed gas was not modified at all. This is demonstrated in Figure 4, where the first part of the experiment shows that the
isotopic composition of $CO_2$ is unaltered when dry sample air is passed through the NAD relative to the direct measurement
of the dry sample air. The second part of the experiment demonstrates that the isotopic composition of $CO_2$ still remained
unaltered when wet sample air is passed through the NAD (thus dried) relative to the direct measurement of the dry sample
air. Since the TILDAS-SICAS is not designed for continuous measurements, we performed this experiment at a lower flow
rate than what would otherwise be used in field to obtain more discrete measurements while processing a certain volume of
air. This demonstrates that, even with a doubling of residence time in the NAD compared to field conditions, the isotopic
composition remains unaltered.

A comprehensive experiment was performed to test the NAD in combination with the PFP-PCP sampling platform at
NOAA-ESRL and INSTAAR, where all the atmospheric trace gases were measured as they will be done in Brazil. These
experiments were performed in four different conditions and the results are summarized in Figure 6. The results show that
for most species, e.g. CO, $CH_4$, $N_2O$ and $SF_6$, the measurements are unaffected when the NAD was used for drying the
sampling air and were within or very close, in the case of $N_2O$, to the WMO network compatibility goals. Even in the case of
$CO_2$, the mole fraction measurements were not severely affected and stayed within the WMO network compatibility goals.
As expected, the isotopic composition of $^{18}O$ in $CO_2$ was affected in the cases where wet samples were collected relative to
dry sample air or wet air dried with the NAD. Additionally, the isotopic composition of $^{13}C$ in $CO_2$ also remained unaffected
in these test conditions.

Through these results presented in this manuscript, we show that the NAD is a viable drying solution and can be used during
flight sampling. The NAD, as shown here, does not affect the composition of the whole air samples, with respect to the
species described in this manuscript and also doesn't affect the isotopic composition of $CO_2$.

*Data availability*. The datasets are available upon request from the corresponding authors.



***Authors contributions***. DP, HAS, and WP designed the setup with suggestions from JBM, AMC and HAJM. HAS, DP, HGJ, BAMK, SEM performed the experiments. DP and HAS analyzed the data. LG, LD, CC and RL helped making the NAD compatible with the PCP-PFP system and its deployment in the sampling sites. DP, HAS, and WP wrote the manuscript with inputs from all the coauthors.

***Competing interests***. The authors declare that they have no conflict of interest.

***Acknowledgements***. We greatly acknowledge the collaboration and kind support of the NOAA Earth System Research Laboratory (ESRL) in Boulder, and the Institute of Arctic and Alpine Research (INSTAAR), University of Colorado, Boulder. In particular we are grateful for both the help and support of Jack Higgs, Don Neff, and Colm Sweeney at NOAA,
and Bruce Vaugh and colleagues at INSTAAR, and for providing their laboratories for our experiments. This work was funded under the H2020 project ASICA (CoG 649087) from the European Research Council.

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
