# Peer review of "Evaluation of a field-deployable NafionTM-based air drying system for collecting whole air samples and its application to stable isotope measurements of CO2"

_Atmospheric Measurement Techniques, 2019_

## Referee Comment (RC1) · Anonymous Referee #1 · 15 Oct 2019

Scientific Significance: 1 The authors have prepared an important discussion paper examining the drying of air sampled into flasks packages that fits well within the scope of AMT papers. This Nafion-based air sample Dryer (NAD) method provides an efficient, effective and logistically practical, method to dry air samples without effecting either the mole fractions or the stable isotopic composition of the air sampled.

Scientific quality: 1 The authors examine parameters required to insure the integrity of the samples, and then a NAD develop a drying system that meets these requirements based on multi-tube Nafion driers. The methods used are sound with a structure to the

examination of the drying system that provides confidence that the authors are fully aware of effects that exist with other drying systems. The examination of both mole fractions and isotopic composition in a very controlled manner with techniques that are proven ensures that their results are sound. The assessment of the NAD system under conditions that closely match the real sampling environments additionally provides confidence that the experimental results are meaningful. While the assessment of the isotopic composition in section 3.3 was at a reduced flow rate compared to both normal operation and the assessment in section 3.4, the authors indicate that the lower flow rate will impact more heavily as the residence time is greater. The authors may wish to discuss the results from section 3.3 and the case 5 results from section 3.4 which are similar tests at different flow rates to demonstrate that their assertion is correct.

Presentation quality: 2 The authors provide experiments that demonstrate the ability of the NAD to sample air with minimal alteration, they compare their results to the WMO GAW compatibility goals. It should be noted that while the flask sampling on aircraft takes place in Brazil the compatibility goal for CO2 should be 0.05 ppm for the Southern Hemisphere.

In section 2.1 the authors describe the use of a G2301 cavity ring down spectrometer which measure CO2, CH4 and H2O. They do not provide a measurement precision for H2O, but rather for carbon monoxide which is not measured by the instrument.

A molecular sieve type 3A is employed to dry the backflush air for the Nafion, the authors may wish to provide manufacturer and grade details in section 2.2 line 140.

The authors have prepared a well structured and readable manuscript. There are several typographical errors that if resolved would improve the paper. Some examples of these are: Page 2 ln35, missing word after that. "We estimated that least 8 flasks . . ." Page 9 ln 223, A comma is required after "each" in the text. "24-inch Perma Pure PD-Series gas dryers containing 200 Nafion tubes each in a stainless steel tube shell"

The GAW report 242 should include the names of the editors in the reference.

Figure 2. The scale for H2O % needs some attention between 0.1 and 1 as the decimal place is not present.

Figure 5. The authors refer to the excess flow line within the text and state that the CRDS instrument and the hygrometer are both attached to this line. In the Figure 5 this is depicted as two separate lines. The authors may wish to clarify either the text or the figure to indicate clearly the configuration used.

---

## Referee Comment (RC2) · Anonymous Referee #2 · 8 Apr 2020

1. Does the paper address relevant scientific questions within the scope of AMT? Yes. 2. Does the paper present novel concepts, ideas, tools, or data? Yes. 3. Are substantial conclusions reached? Yes. 4. Are the scientific methods and assumptions valid and clearly outlined? Yes. 5. Are the results sufficient to support the interpretations and conclusions? Yes. 6. Is the description of experiments and calculations sufficiently complete and precise to allow their reproduction by fellow scientists (traceability of results)? Yes 7. Do the authors give proper credit to related work and clearly indicate their own new/original contribution? Yes. 8. Does the title clearly reflect the contents of

the paper? Yes. 9. Does the abstract provide a concise and complete summary? Yes. 10. Is the overall presentation well-structured and clear? Largely, yes. See 13. below. 11. Is the language fluent and precise? To large extent yes, but improvements are possible. Check for typos and missing punctuation. 12. Are mathematical formulae, symbols, abbreviations, and units correctly defined and used? Yes. 13. Should any parts of the paper (text, formulae, figures, tables) be clarified, reduced, combined, or eliminated? There are some repetitions in the text that may be omitted without losing information. Figure captions are too lengthy. 14. Are the number and quality of references appropriate? Yes – with the exception that I would welcome seeing a reference or two more on Nafion usage for CO2 measurements (and possible uncertainties related to this). 15. Is the amount and quality of supplementary material appropriate? Not applicable.

General comments

The authors are presenting their work on a Nafion-based air drying system for collecting whole air samples and its application to stable isotope measurements of CO2. This is a contribution well in scope of AMT. It describes equipment and procedures that will prove useful to everyone who aims for taking air samples for high quality measurements of CO2 and other trace gases in challenging environments. The analytical approach and validation methods are sound and well documented. If some minor shortcomings are addressed (see a separate file named "amt-2019-295-supplement.pdf"), I recommend publication of this manuscript in AMT. Data from the manuscript should be available in a freely accessible repository or as supplement to the paper, not upon request with the authors.

Specific comments and technical corrections

They are included in a separate file named "amt-2019-295-supplement.pdf".

Please also note the supplement to this comment:

https://www.atmos-meas-tech-discuss.net/amt-2019-295/amt-2019-295-RC2-supplement.pdf

**Supplement:**

[Figure]

[Figure]

**Evaluation of a field-deployable Nafion-based air drying system for collecting whole air samples and its application to stable isotope measurements of CO₂**

Dipayan Paul[1,*], Hubertus A. Scheeren[1,*], Henk G. Jansen[1], Bert A. M. Kers[1], John B. Miller[2], Andrew M. Crotwell[2,3], Sylvia

5   E. Michel[4], Luciana V. Gatti[5], Lucas G. Domingues[5], Caio S. C. Correia[5], Raiane A. L. Neves[5], Harro A. J. Meijer[1], Wouter Peters[1,6]

[1]*Centre for Isotope Research (CIO), University of Groningen, Groningen, the Netherlands*
[2]*National Oceanic and Atmospheric Administration (NOAA), Earth System Research Laboratory, Boulder, Colorado, USA*
[3]*Cooperative Institute for Research in Environmental Sciences (CIRES), University of Colorado, Boulder, Colorado, USA*
10  [4]*Institute of Arctic and Alpine Research (INSTAAR), University of Colorado, Boulder, Colorado, USA*
[5]*National Institute of Space research (INPE), Atmospheric Chemistry Department, San Jose dos Campos, Brazil*
[6]*Department of Meteorology and Air Quality, Environmental Sciences Group, Wageningen University, the Netherland*

*both authors contributed equally.*

15  *Correspondence to: Dipayan Paul (d.paul@rug.nl) and Hubertus A. Scheeren (h.a.scheeren@rug.nl)*

**Abstract.** Atmospheric flask samples are either collected at atmospheric pressure by simply opening a valve of a pre-evacuated flask, or pressurized with the help of a pump to a few bar above ambient providing large air samples for analysis. Under humid conditions, there is a risk that water vapour in the sample leads to condensation on the walls of the flask, notably at higher than ambient sampling pressures. Liquid water in sample flasks is known to affect the $CO_2$ mixing ratios

20  and also alters the isotopic composition of oxygen ($^{17}O$ and $^{18}O$) in $CO_2$ via isotopic equilibration. Hence, for accurate determination of $CO_2$ mole fractions and its stable isotopic composition, it is vital to dry the air samples to a sufficiently low dew point before they are pressurized in flasks to avoid condensation. Moreover, the drying system itself should not influence the mixing ratio and the isotopic composition of $CO_2$, nor of the other constituents under study. For the "Airborne Stable Isotopes of Carbon from the Amazon" (ASICA) project focusing on accurate measurements of $CO_2$ and its singly-

25  substituted stable isotopologues over the Amazon, an air drying system was needed capable of removing water vapour from air sampled at a dew point below than −2 °C, high flow rates up to 12 L/min, and without the need for electrical power. Since to date, no commercial air drying device is available that meets these requirements, we designed and built our own consumable-free, power-free, and portable drying system based on multi-tube Nafion™ gas sample driers (Perma Pure, Lakewood, USA). The required dry purge air is provided by feeding the exhaust flow of the flasks sampling system through

30  a dry molecular sieve (type 3A) cartridge. In this study we describe the systematic evaluation of our Nafion-based air sample dryer with emphasis on its performance concerning the measurements of atmospheric $CO_2$ mole fractions and the three singly-substituted isotopologues of $CO_2$ ($^{16}O^{13}C^{16}O$, $^{16}O^{12}C^{17}O$ and $^{16}O^{12}C^{18}O$), as well as the trace gas species $CH_4$, $CO$,

**Summary of Comments on amt-2019-295_RC2.pdf**

**Page: 1**

Number: 1 Author: reviewer 2 Subject: Cross-Out Date: 07/04/2020 13:31:49

Number: 2 Author: reviewer 2 Subject: Sticky Note Date: 07/04/2020 13:38:21
ambient pressure.

Number: 3 Author: reviewer 2 Subject: Cross-Out Date: 07/04/2020 13:32:50

Number: 4 Author: reviewer 2 Subject: Sticky Note Date: 07/04/2020 13:38:46
lower

Number: 5 Author: reviewer 2 Subject: Cross-Out Date: 07/04/2020 13:43:11

$N_2O$, and $SF_6$. Experimental results simulating extreme tropical conditions (saturated air at 33 °C) indicated that the response of the air dryer is almost instantaneous and that approximately 85 L of air, containing up to 4% water vapour, can be

35 processed staying below a −2 °C dew point temperature (at 275 kPa). We estimated that least 8 flasks can be sampled (at an overpressure of 275 kPa) with a water vapour content below −2 °C dew point temperature during a typical flight sampling up to 5 km altitude over the Amazon, whereas the remaining samples would stay well below 5 °C dew point temperature (at 275 kPa). The performance of the air dryer on measurements of $CO_2$, $CH_4$, $CO$, $N_2O$, and $SF_6$, and the $CO_2$ isotopologues $^{16}O^{13}C^{16}O$ and $^{16}O^{12}C^{18}O$ was tested in the laboratory simulating real sampling conditions by compressing humidified air

40 from a calibrated cylinder, after being dried by the air dryer, into sample flasks. We found that the mole fraction and the isotopic composition difference between the different test conditions (including the dryer) and the base condition (dry air, without dryer) remained well within or very close to, in the case of $N_2O$, the WMO recommended compatibility goals for independent measurement programs, proving that the test condition induced no significant bias on the sample measurements.

**1 Introduction**

45 $CO_2$ [1] is one of the most important and extensively monitored greenhouse gases in the atmosphere. Atmospheric $CO_2$ [2] mole fraction measurements provide information that helps understand the continuously increasing mole fractions in the atmosphere due to the combination of human activities, and exchange with the terrestrial- and oceanic components of the global carbon cycle. [4] Further, measurements of the isotopic composition of the atmospheric $CO_2$ provides [3] information about the [6] sources [5] and sinks. $CO_2$ mole fraction can either be continuously measured using instruments capable of performing

50 continuous-flow measurements in whole air samples e.g., using nondispersive infrared (NDIR) based sensors (Stephens et al., 2011), cavity ring-down spectrometers (Chen et al., 2010) or quasi-continuously by using gas chromatography (van der Laan et al., 2009). Alternatively, discrete air samples can be collected in flasks for later analysis in a laboratory. Flasks are typically filled with ambient air either by opening the valve of a pre-evacuated flask, or by pressurizing a flask with the help of a pump. Under humid conditions, flask sampling requires drying of the sample air to prevent condensation inside the flask

55 which can affect the $CO_2$ mole fractions as well as for the oxygen stable isotopes composition (Gemery et al., 1996; Trolier et al., 1996).

Since 2009, a substantial effort is undertaken to establish a long-term atmospheric mole fraction $CO_2$ record over the Amazon rain forest. Air samples are collected onboard a small aircraft along a vertical profile from 4.4 km down to 300 m amsl (above mean sea level) at a bi-monthly rate at [7] fferent sites (Alta Floresta (ALF), Rio Branco (RBA), Santarém

60 (SAN), and Tefé (TEF)) over the Amazon forest. Additionally, samples are also collected once every month at Manaus (MAN) and over a big flooded area in a different ecosystem at Pantanal (PAN, Mato Grosso state). This unique $CO_2$ program resulted in a number of new insights on *net* carbon exchange from this region (Alden et al., 2016; Gatti et al., 2014; van der Laan-Luijkx et al., 2015) and the measurements continue still. The project "Airborne Stable Isotopes of Carbon from

**Page: 2**

| | | | |
|---|---|---|---|
| Number: 1 | Author: reviewer 2 | Subject: Sticky Note | Date: 07/04/2020 13:47:08 |

Write out the first time used.

| | | | |
|---|---|---|---|
| Number: 2 | Author: reviewer 2 | Subject: Sticky Note | Date: 07/04/2020 13:47:51 |

dry air mole fractions

| | | |
|---|---|---|
| Number: 3 | Author: reviewer 2 | Subject: Cross-Out  Date: 07/04/2020 13:49:09 |

| | | |
|---|---|---|
| Number: 4 | Author: reviewer 2 | Subject: Cross-Out  Date: 07/04/2020 13:48:58 |

| | | | |
|---|---|---|---|
| Number: 5 | Author: reviewer 2 | Subject: Sticky Note | Date: 07/04/2020 13:49:43 |

its

| | | |
|---|---|---|
| Number: 6 | Author: reviewer 2 | Subject: Cross-Out  Date: 07/04/2020 13:49:36 |

| | | | |
|---|---|---|---|
| Number: 7 | Author: reviewer 2 | Subject: Sticky Note | Date: 07/04/2020 13:52:28 |

four

[revised manuscript text omitted]

Number: 1       Author: reviewer 2   Subject: Inserted Text       Date: 07/04/2020 14:46:04

-

Number: 2       Author: reviewer 2   Subject: Cross-Out   Date: 07/04/2020 14:46:58

Number: 3       Author: reviewer 2   Subject: Cross-Out   Date: 07/04/2020 14:47:03

Number: 4       Author: reviewer 2   Subject: Sticky Note       Date: 07/04/2020 14:48:17

chose?

Number: 5       Author: reviewer 2   Subject: Inserted Text       Date: 07/04/2020 14:47:11

,

Number: 6       Author: reviewer 2   Subject: Sticky Note       Date: 07/04/2020 14:49:30

you should add a statement about if and how the 3A mol. sieve works/alters the composition of the gas.

Number: 7       Author: reviewer 2   Subject: Inserted Text       Date: 07/04/2020 14:50:18

,

Number: 8       Author: reviewer 2   Subject: Cross-Out   Date: 07/04/2020 14:52:54

Number: 9       Author: reviewer 2   Subject: Cross-Out   Date: 07/04/2020 14:51:31

Number: 10       Author: reviewer 2   Subject: Sticky Note       Date: 07/04/2020 14:54:15

As a single...... would therefore not be...., we decided to use two of them to increase....

Number: 11       Author: reviewer 2   Subject: Sticky Note       Date: 07/04/2020 14:55:09

- 2 or +2?

[Figure]

255   this manuscript, most samples collected in the Amazonia have a water vapour content lower than −2 °C dew point (~60 % RH) and none would exceed the 5 °C dew point (~100 % RH) limit at 275 kPa flask pressure. In summary, for an optimal performance under tropical saturated conditions we decided to use the PD-200T with a total length of 48" (~140 cm). For practical reasons we chose to use two of the 24" (~70 cm) long PD-200T-24MSS, which can be placed in series (in a parallel configuration) to save space.

260   The amount of molecular sieve needed to sufficiently dry the purge gas was initially set on ~150 g. From our experiments testing the drying capacity of the double Nafion tube we found that a minimum of 175 g of dry molecular sieve was needed for drying a minimum of 200 L of air at moderate conditions of ~22 °C and 70 % RH down to <1 % RH. For a safe margin with respect to the humid tropical conditions we meet over the Amazon region we decided to double the amount of molecular sieve to 350 g. Our work on testing the drying capacity of the NAD is further elaborated in the next section.

265   ### 3.2 Drying capacity of the NAD

The drying capacity of Nafion is dependent on the dryness of the purge flow and the individual Nafion tubes. The hydrophilic properties of sulfonated tetrafluoroethylene polymer, of which Nafion is formed, absorb water molecules at the humid side and releases it at the dry side creating a permeation of water from the wet-side to the dry-side. When the Nafion material is not properly dried before usage, the material will remain saturated with water molecules for a longer period of
270   time and will thus limit its total capability to dry the sample air stream. Hence, before each experiment or operation with the NAD it was essential to dry the Nafion tubes by flushing them with dry compressed air at a rate of 2 L/min at STP for about 12 hours. After this time, the outflow of the NAD would show no remaining water vapour, i.e. the outflow equals the inflow water vapour content indicating the Nafion is dry.

Here we describe the results of a specific experiment where we mimicked the sample collection process during a flight under
275   tropical conditions to demonstrate the usability and the performance of the NAD. This experiment also allowed us to determine the total amount of water the NAD was capable of removing in its current set-up. The experiments were performed in a greenhouse test facility at the biology department of the University of Groningen where the water content of the sampled air could be set to 3.7-3.9 % at ~33 °C mean temperature. As described earlier in section 3.1, a sampling sequence using the PCP-PFP system consist of 5 L of air for flushing the PCP-PFP inlet and manifold, followed by flushing
280   the sample flask with 10 L of air, and ending with pressurizing the sample flask up to 275 kPa before closing the upstream valve, corresponding to ~1.9 L of air at STP in the flask. The complete filling of a flask was simulated by flushing the NAD periodically at 6 L/min with ambient air from the greenhouse for a duration of 3 minutes, depicted in Figure 2. The bottom left axis in Figure 2 shows the water vapour content, as measured by the Picarro CRDS; the top left axis shows the corresponding dew point temperature at 275 kPa; and the right axis shows the total volume of air processed through the
285   NAD. The first 50 seconds correspond to the inlet and manifold flushing (blue points in the dew point data), the next 100

Number: 1     Author: reviewer 2   Subject: Inserted Text        Date: 07/04/2020 14:58:26

,

Number: 2     Author: reviewer 2   Subject: Inserted Text        Date: 07/04/2020 14:59:04

,

Number: 3     Author: reviewer 2   Subject: Cross-Out  Date: 07/04/2020 15:01:47

[revised manuscript text omitted]

**Page: 13**

Number: 1          Author: reviewer 2   Subject: Cross-Out  Date: 07/04/2020 15:10:32

Number: 2          Author: reviewer 2   Subject: Sticky Note          Date: 07/04/2020 15:13:47
please comment/explain why the test is still valid even if you did not go to ~ 4 %vol H2O

360 processed in the dry mode and wet mode (including the stabilization period) was 162 L and 174 L, respectively, at a flow
rate of 4.5 L/min. As can be seen from the data in Figure 4, the isotopic composition of $CO_2$ in the cylinder air is not altered
by the NAD, both in dry and wet conditions. One sample measurement (measurement number 16) in the dry mode was
affected by an unknown cause and was considered an outlier, thus not included in the analysis. The mean of the ZE
measurements (± SE), corresponding to the dry and wet mode, for the three singly-substituted isotopologues of $CO_2$ are also

365 shown in Figure 4. We note that the flow rate used during this experiment (4.5 L/min) is significantly lower than the flow
rates typically used during sample collection (~12 L/min). This clearly demonstrates that under laboratory test conditions the
NAD has negligible effect on the isotopic composition of $CO_2$, ~~even with significantly longer residence times in the Nafion
tubes.~~

[Figure]

370 **Figure 4**. *A zero-enrichment experiment performed with the TILDAS-SICAS to test the effect of the NAD on the $CO_2$ mole
fraction and the stable isotopes of $CO_2$. The top panel shows the $CO_2$ mole fraction, where the grey triangles represent the
sample measurements and the black inverted triangles represent the reference measurements. The other three panels show
the isotopic composition of the sample gas determined with respect to the reference gas. The reference gas is identical to the
sample gas, the only difference being that the sample gas was measured after it had passed through the NAD. The first part*

375 *of the experiment represents dry mode (shown in red background) where, for sample measurements, dry air from a cylinder
was passed through a preconditioned NAD and was then introduced into the TILDAS-SICAS for measurements. In the latter*

Number: 1    Author: reviewer 2   Subject: Sticky Note    Date: 07/04/2020 15:34:43

Number: 2    Author: reviewer 2   Subject: Sticky Note    Date: 07/04/2020 15:20:32
true for lab test/conditions - but this does not explicitly confirm that at ~12L/min this still holds - has to be demonstrated/discussed

Number: 3    Author: reviewer 2   Subject: Cross-Out  Date: 07/04/2020 15:19:31

Number: 4    Author: reviewer 2   Subject: Inserted Text    Date: 07/04/2020 15:21:42
dry air

[revised manuscript text omitted]

Number: 1     Author: reviewer 2   Subject: Sticky Note     Date: 07/04/2020 15:30:18
try to avoid doubling of text - in text or in caption only is sufficient - lengthy explanations are better located in text

Number: 2     Author: reviewer 2   Subject: Sticky Note     Date: 07/04/2020 15:40:38
which compatibility goals (I know it is the WMO goals, but this is not clear from this portion of the text)

Number: 3     Author: reviewer 2   Subject: Sticky Note     Date: 07/04/2020 15:39:26
This may be, but this does not in itself mean that the performance of these two systems is equal. Please elaborate a bit more - make it clearer what you want to say when mentioning this.

[revised manuscript text omitted]

Number: 1     Author: reviewer 2   Subject: Sticky Note     Date: 08/04/2020 00:26:57
I am missing just a bit more detail (independent from the manufacturer information) on the influence of (humid) nafion membrane on the CO2 concentration of the sampled air.

Number: 2     Author: reviewer 2   Subject: Inserted Text     Date: 07/04/2020 23:46:55
,

Number: 3     Author: reviewer 2   Subject: Inserted Text     Date: 07/04/2020 23:46:48
-

[Figure]

NAD which is configured for use with the PCP-PFP system from NOAA-ESRL, although the use of our system is not limited to that sampling platform.

During the development phase we learned that to achieve the best performance from the dryers in the configuration we wanted to use, they had to be conditioned before every use. Conditioning was performed by flushing the Nafion drying tubes and the purge volume overnight with dry air at ~2 L/min, and a successful conditioning was achieved when the water content in the dry air entering the dryer was equal to the water content in the air exiting the dryer. Similarly, the molecular sieve granules were also dried following every use by baking them overnight in an oven set at 100 °C. Since all the end connectors on the NAD are normally-closed Swagelok Quick-Connect connectors, the system is filled with dry air at ambient pressure and stored. We performed a storage stability check over a period of one month and the results indicated that the NAD, if stored in dry conditions, would perform similarly to one freshly conditioned. This property is particularly beneficial for the sampling conditions in Brazil because the conditioning step is performed in the lab few days before the PFP and the NAD are shipped to the sample collection site.

In the ASICA program, the goal is to perform measurements of $CO_2$ mole fraction and high precision measurements of all the singly-substituted isotopologues of $CO_2$ to constrain the gross primary production and its response to droughts for the Amazon basin. In this program, twelve samples are collected per flight at altitudes between 300 m and 4.4 km amsl. To achieve high precision measurements of the isotopic composition of $CO_2$ in the whole air samples, the collected sample air must be dry. Thus the next requirement was to estimate the water removal capacity of the NAD and estimate the number of flasks that could be filled with a targeted dew point of −2 °C (at 275 kPa). From the experiment presented in Figure 2, it is evident that the response of the NAD is almost instantaneous and approximately 85 L of air, containing 3.7-3.9 % $H_2O$, can be processed within the targeted dew point of −2 °C. In Figure 3, we estimated the number of flasks that can be sampled within the targeted dew point temperature during a typical flight above the Amazon. According to these estimates, we can at least collect 8 flasks with water vapour content below −2 °C dew point temperature (at 275 kPa, 60 % RH), whereas the rest of the flasks would still contain water vapour below 5 °C dew point temperature (at 275 kPa, 100 % RH). As discussed earlier, a previous study (Gemery et al., 1996) showed that flask samples containing air with relative humidity below 60% are generally not affected and gradual biases are observed in the 60-100 % RH range. This would thus indicate that the first eight samples collected between 4.4-1.5 km should be free from any bias introduced from the water vapour content in the flask. But the last four samples collected between 1.5-0.3 km could be potentially biased towards the isotopic composition of the water it is exposed to, although this bias should be gradual due to the continuous increase of the water vapour content as more samples are processed through the NAD. This prediction, shown in Figure 3, is of course based on the experiment shown in Figure 2, where the NAD was directly and continuously exposed to air containing ~4 % water vapour. This scenario would saturate the NAD much faster and the ability to dry the Nafion tubes through the purge side would be slower. Whereas, in a real sampling scenario, the NAD is not exposed to high water vapour content as soon as sampling begins and

Number: 1        Author: reviewer 2  Subject: Sticky Note        Date: 07/04/2020 23:55:02

please be more specific

Number: 2        Author: reviewer 2  Subject: Sticky Note        Date: 07/04/2020 23:56:42

please be more specific

Number: 3        Author: reviewer 2  Subject: Sticky Note        Date: 07/04/2020 23:59:01

I think you would do your setup more justice, if you did not limit it here to the utilisation in Brazil - even if it was constructed for this purpose. It can be used in many other settings.

[Figure]

490 thus the purge side could still dry the Nafion tubes longer than in the former case. This would then lead to an increase in the estimated quantity of water removed by the NAD before crossing the 60 % RH limit (−2 °C dew point) in the sampled flasks, as shown in Figure 3, and would thus encompass more than 8 samples within this limit and would also lower the biases for the ones outside the limit.

The next requirement was to establish if the NAD was inert for the gases-of-interest and did not alter the isotopic composition of $CO_2$ while sampling. To understand the effect of the NAD on the isotopic composition of $CO_2$, we performed 495 a semi-continuous zero-enrichment experiment with the TILDAS-SICAS instrument in our laboratory. In such an experiment, the same gas is treated both as a reference and a sample gas, where the reference stream is unprocessed and the sample stream is processed. Thus, a zero-difference between the reference and the sample stream would indicate that the processed gas was not modified at all. This is demonstrated in Figure 4, where the first part of the experiment shows that the isotopic composition of $CO_2$ is unaltered when dry sample air is passed through the NAD relative to the direct measurement 500 of the dry sample air. The second part of the experiment demonstrates that the isotopic composition of $CO_2$ still remained unalt[1] when wet sample air is passed through the NAD (thus dried) relative to the direct measurement of the dry sample air. Since the TILDAS-SICAS is not designed for continuous measurements, we performed this experiment at a lower flow rate than what would otherwise be used in field to obtain more discrete measurements while processing a certain volume of air. This demonstrates that, even with [2] ubling of residence time in the NAD compared to field conditions, the isotopic 505 composition remains unaltered.

A comprehensive experiment was performed to test the NAD in combination with the PFP-PCP sampling platform at NOAA-ESRL and INSTAAR, where all the atmospheric trace gases were measured as they will be done in Brazil. These experiments were performed in four different conditions and the results are summarized in Figure 6. The results show that for most species, e.g. CO, $CH_4$, $N_2O$ and $SF_6$, the measurements are unaffected when the NAD was used for drying the 510 sampling air and were within or very close, in the case of $N_2O$, to the WMO network compatibility goals. Even in the case of $CO_2$, the mole fraction measurements were not severely affected and stayed within the WMO network compatibility goals. As expected, the isotopic composition of $^{18}O$ in $CO_2$ was affected in the cases where wet samples were collected relative to dry sample air or wet air dried with the NAD. Additionally, the isotopic composition of $^{13}C$ in $CO_2$ also remained unaffected in these test conditions.

515 Through[3] these results presented in this manuscript, we show that the NAD is a viable drying solution and can be used during flight sampling. The NAD, as shown here, does not affect the composition of the whole air samples, with respect to the species described in this manuscript and also doesn[4] affect the isotopic composition of $CO_2$.

***Data availability***. The datasets[5] available upon request from the corresponding authors.

Number: 1        Author: reviewer 2   Subject: Sticky Note       Date: 08/04/2020 00:07:58

There are differences, albeit small - therefore rather, for example, the differences between the different experiments are in the range of measurement uncertainty

Number: 2        Author: reviewer 2   Subject: Sticky Note       Date: 08/04/2020 00:12:37

It would make a more complete discussion if you elaborated in a bit more detail on this point (i.e. why is a shorter residence time even more favorable, what are the processes involved, surface processes, kinetic fractionation, etc.)

Number: 3        Author: reviewer 2   Subject: Cross-Out   Date: 08/04/2020 00:14:01

Number: 4        Author: reviewer 2   Subject: Sticky Note       Date: 08/04/2020 00:14:21

does not

Number: 5        Author: reviewer 2   Subject: Sticky Note       Date: 08/04/2020 00:16:32

I am unsure about the AMT policy this regarding, but my conviction is that all data used in a paper have to be freely available/deposited in a repository which is freely accessible, independently of the authors.

[revised manuscript text omitted]

Number: 1          Author: reviewer 2   Subject: Sticky Note          Date: 08/04/2020 00:29:06
update reference

---

## Author Comment (AC1) · 20 May 2020

**Authors response to anonymous referee #1 on "Evaluation of a field-deployable Nafion™-based air drying system for collecting whole air samples and its application to stable isotope measurements of CO₂" by Paul, D. et al.**

Dear Referee,

Thanks a lot for your valuable and constructive comments. We have revised our manuscript based on the comments we received. Through this document, we are addressing all comments we received, shown in *italic* font and our responses to them are shown in regular font (inserted texts are underlined).

In addition, there were some errors in the annotations of Figure 4 which has been updated (n in dry mode was 11 and not 12; standard error of the mean corresponding to $\delta^{13}C$ and $\delta^{18}O$ was changed to 0.013 and 0.005, respectively from 0.008 and 0.004).

Sincerely,

Dipayan

(on behalf of all co-authors)

- *Scientific quality: 1 The authors examine parameters required to insure the integrity of the samples, and then a NAD develop a drying system that meets these requirements based on multi-tube Nafion driers. The methods used are sound with a structure to the examination of the drying system that provides confidence that the authors are fully aware of effects that exist with other drying systems. The examination of both mole fractions and isotopic composition in a very controlled manner with techniques that are proven ensures that their results are sound. The assessment of the NAD system under conditions that closely match the real sampling environments additionally provides confidence that the experimental results are meaningful. While the assessment of the isotopic composition in section 3.3 was at a reduced flow rate compared to both normal operation and the assessment in section 3.4, the authors indicate that the lower flow rate will impact more heavily as the residence time is greater. The authors may wish to discuss the results from section 3.3 and the case 5 results from section 3.4 which are similar tests at different flow rates to demonstrate that their assertion is correct.*

Authors response: Indeed, experiment shown in Figure 4 is comparable to the results shown in Figure 6 (case 5). We have added the following text at the end of section 3.4.

"... Case 5 is in fact comparable to the experiment shown in Figure 4, only differing in their used flow rates and that the former being a flow-through semi-continuous measurement scheme. Although we have argued that higher flow rates are likely favourable for reduced isotopic exchange (observable in $\delta^{18}O$) due to the reduction in the interaction time between the NAD surface and CO₂, Case 5 is slightly more biased than expected, based on Figure 4. This is likely caused by the additional and variable interaction of the sample with the flask surface, not encountered during the flow-through experiment shown in Figure 4."

- *Presentation quality: 2 The authors provide experiments that demonstrate the ability of the NAD to sample air with minimal alteration, they compare their results to the WMO GAW compatibility goals. It should be noted that while the flask sampling on aircraft takes place in Brazil the compatibility goal for CO2 should be 0.05 ppm for the Southern Hemisphere.*

Authors response: We have added the Southern Hemisphere compatibility goals in Figure 6.

- *In section 2.1 the authors describe the use of a G2301 cavity ring down spectrometer which measure CO2, CH4 and H2O. They do not provide a measurement precision for H2O, but rather for carbon monoxide which is not measured by the instrument.*

Authors response: We have corrected this part and reads as follows:

"…The overall measurement precision of the CRDS-systems used was typically <0.03 µmol mol$^{-1}$ (ppm) for $CO_2$, <0.2 nmol mol$^{-1}$ (ppb) for $CH_4$, based on our long-term measurements of target cylinders, and <30 ppm for H2O, based on manufacturers specifications."

- *A molecular sieve type 3A is employed to dry the backflush air for the Nafion, the authors may wish to provide manufacturer and grade details in section 2.2 line 140.*

Authors response: Added.

"The NAD contains two Perma Pure PD-Series™ Nafion™ dryers (PD-200T-24-MSS), a molecular sieve cartridge (type 3A, ~2 mm beads, 350 g, Sigma Aldrich), a 2 micron in-line filter (Swagelok, SS-4FW-2), stainless steel tubing and various Swagelok connectors."

- *The authors have prepared a well structured and readable manuscript. There are several typographical errors that if resolved would improve the paper. Some examples of these are: Page 2 ln35, missing word after that. "We estimated that least 8 flasks ..." Page 9 ln 223, A comma is required after "each" in the text. "24-inch Perma Pure PD-Series gas dryers containing 200 Nafion tubes each in a stainless steel tube shell"*

Authors response: We have corrected these sentences:

"…We estimated that at least 8 flasks can be sampled (at an overpressure of 275 kPa) with a water vapour content below −2 °C dew point temperature during a typical flight sampling up to 5 km altitude over the Amazon, whereas the remaining samples would stay well below 5 °C dew point temperature (at 275 kPa)."

"…Due to the relatively high flow rate of the PCP-PFP sampling system of up to 15 L/min we choose to use the 24-inch Perma Pure PD-Series gas dryers containing 200 Nafion™ tubes each, in a stainless steel tube shell designed for high flows up 40 L/min."

- *The GAW report 242 should include the names of the editors in the reference.*

Authors response: This reference has been modified to:

19th WMO/IAEA Meeting on Carbon Dioxide, Other Greenhouse Gases and Related Measurement Techniques (GGMT-2017), 27-31 August 2017, Dübendorf, Switzerland, Edited by Andrew Crotwell and Martin Steinbacher, GAW Report No. 242, 2017.

- *Figure 2. The scale for H2O % needs some attention between 0.1 and 1 as the decimal place is not present.*

Authors response: The figure has been updated:

[Figure]

- *Figure 5. The authors refer to the excess flow line within the text and state that the CRDS instrument and the hygrometer are both attached to this line. In the Figure 5 this is depicted as two separate lines. The authors may wish to clarify either the text or the figure to indicate clearly the configuration used.*

Authors response: The schematic shown in Figure 5 depicts the correct configuration and the text has been adjusted accordingly.

"A Picarro CRDS was used to measure the stability of $CO_2$, $CH_4$, and $H_2O$ in the flow exiting the buffer volume, as shown in Figure 5."

---

## Author Comment (AC2) · 20 May 2020

**Authors response to anonymous referee #2 on "Evaluation of a field-deployable Nafion™-based air drying system for collecting whole air samples and its application to stable isotope measurements of CO₂" by Paul, D. et al.**

Dear Referee,

Thanks a lot for your valuable and constructive comments. We have revised our manuscript based on the comments we received. Through this document, we are addressing all comments we received, shown in *italic* font and our responses to them are shown in regular font (inserted texts are underlined).

In addition, there were some errors in the annotations of Figure 4 which has been updated (n in dry mode was 11 and not 12; standard error of the mean corresponding to $\delta^{13}C$ and $\delta^{18}O$ was changed to 0.013 and 0.005, respectively from 0.008 and 0.004).

Sincerely,

Dipayan

(on behalf of all co-authors)

Specific comments and technical corrections from https://www.atmos-meas-tech-discuss.net/amt-2019-295/amt-2019-295-RC2-supplement.pdf

- *Referee's comments on page 1:*

Authors response: All textual suggestions have been incorporated.

- *Referee's comments on page 2:*

Authors response: All textual suggestions have been incorporated.

- *Referee's comments on page 3:*

Authors response: All textual suggestions have been incorporated.

Comment #5: *use TM throughout ms*

Authors response: "Nafion" has been changed to "Nafion™" throughout the manuscript.

- *Referee's comments on page 4:*

Authors response: All textual suggestions have been incorporated.

Comment #1: *define all acronyms at first use*

Authors response: We have now expanded all the acronyms.

Comment #2: *Based on your own measurements?*

Authors response: To address your question regarding the source of the indicated measurement precision, we have added the following text for clarification. Additionally, we have also corrected our text, as pointed

out by Referee #1, which now indicates the measurement precision of $CO_2$, $CH_4$ and $H_2O$ (and not CO, which Picarro G2301 doesn't measure).

"…The overall measurement precision of the CRDS-systems used was typically <0.0$\underline{3}$ µmol mol$^{-1}$ (ppm) for $CO_2$, <0.2 nmol mol$^{-1}$ (ppb) for $CH_4$, based on our long-term measurements of target cylinders, and <30 ppm for $H_2O$, based on manufacturers specifications."

- *Referee's comments on page 5:*

Authors response: All textual suggestions have been incorporated.

Comment #2-3: *type?; you should have all the relevant components listed, with model etc. - best in a Table*

Authors response: We have added all the relevant component part numbers within the text.

- *Referee's comments on page 6:*

Authors response: All textual suggestions have been incorporated.

- *Referee's comments on page 7:*

Authors response: All textual suggestions have been incorporated.

- *Referee's comments on page 8:*

Authors response: All textual suggestions have been incorporated.

Comment #1: *per flight/sampling/....*

Authors response: The sentence has been rephrased to: "For the ASICA project, typically 12 flasks are filled with dried air during each flight sampling."

Comment #2: *do you mention somewhere if this is absolute or above ambient?*

Authors response: We have inserted the word "absolute" to make the filling pressure explicit and the sentence reads as: "A sample is collected by closing the downstream flask valve and pressurizing the flask to 275 kPa (absolute) before closing the upstream valve (corresponding to ~1.9 L of air at STP)."

Comment #4: *it would be good if you could cite a reference here*

Authors response: During the initial stages of this work, we tested $Mg(ClO_4)_2$ as a desiccant and its effect specifically on the stable isotopic composition of $CO_2$. We did not observe any significant deviation in $CO_2$ mole fraction and its stable isotope composition ($\delta^{13}C$ and $\delta^{18}O$) caused by $Mg(ClO_4)_2$.

Comment #5: *not correct, it is a strong oxidizer - it supports combustion*

Authors response: This sentence has been rephrased to: "Perchlorates are stable at normal temperatures, but when they are exposed to high temperatures e.g. in case of a fire, they accelerate combustion."

- *Referee's comments on page 9:*

Authors response: All textual suggestions have been incorporated.

Comment #6: *you should add a statement about if and how the 3A mol. sieve works/alters the composition of the gas.*

Authors response: We have added the following text based on our own experience: "Although an excellent desiccant by itself, molecular sieve (type 3A) cannot be used to directly dry sample air as it tends to alters the composition of air. Hence, we chose to use molecular sieve (type 3A) as a drying agent in the purge flow line because it is additionally non-toxic, economical, and reusable."

Comment #11: *- 2 or +2?*

Authors response: +2°C is correct.

- *Referee's comments on page 10:*

Authors response: All textual suggestions have been incorporated.

- *Referee's comments on page 11:*

Authors response: All textual suggestions have been incorporated.

- *Referee's comments on page 13:*

Authors response: All textual suggestions have been incorporated.

Comment #2: *please comment/explain why the test is still valid even if you did not go to ~ 4 %vol H2O*

Authors response: We have added the following text "… Although the humidity level achieved in these experiments were less than the maximum one would encounter in the Brazilian Amazon (0-3 km), it clearly demonstrates the lack of isotopic exchange caused by the interaction of $CO_2$ with the oxygen-rich Nafion™ surface in the dry mode and in a relatively less severe wet mode. Indeed, further experiments with sample air saturated with water vapour, up to ~4 %, would be needed to confirm a complete lack of isotopic exchange even at high-humidity conditions."

- *Referee's comments on page 14:*

Authors response: All textual suggestions have been incorporated.

Comment #2: *true for lab test/conditions - but this does not explicitly confirm that at ~12L/min this still holds - has to be demonstrated/discussed*

Authors response: We agree with this comment that the presented experiment does not explicitly show that the effect on the stable isotopic composition of $CO_2$ is insignificant even at 12 L/min. The reason this experiment was performed at a flow rate significantly lower than 12 L/min was to measure more discrete samples while processing ~300 L of air: had the experiment been performed at 12 L/min, the experiment would have yielded only 4 isotopic values in each mode (dry and wet) compared to 11 in the presented experiment. Additionally, we also expect that the extended residence time in the NAD (at 4.5 L/min) would allow more time for interaction of $CO_2$ with the NAD surface and thus introducing larger biases.

We have hence added the following sentence: "This clearly demonstrates that under laboratory test conditions the NAD has negligible effect on the isotopic composition of $CO_2$, even with significantly longer residence times in the Nafion™ tubes. It is thus expected that at higher flow rates (12 L/min), the reduced interaction time between the air stream and the NAD surface should have even lesser influence on the isotopic composition of $CO_2$."

- *Referee's comments on page 15:*

Authors response: All textual suggestions have been incorporated.

- *Referee's comments on page 16:*

Authors response: All textual suggestions have been incorporated.

Comment #2: *which compatibility goals (I know it is the WMO goals, but this is not clear from this portion of the text)*

Authors response: We have revised this part and the text reads as follows: "In these experiments, we tested 4 different conditions by filling a set of three flasks under the following conditions: (A) dry air-without dryer, (B) dry air-with dryer, (C) wet air-without dryer, and (D). wet air-with dryer. When the difference between the base condition and the test condition remained within the WMO recommended compatibility goals ($\pm$0.1 and 0.05 ppm for $CO_2$ for the Northern and Southern Hemisphere, respectively; $\pm$2 ppb both for $CH_4$ and CO; $\pm$0.1 ppb for $N_2O$; $\pm$0.02 ppt for $SF_6$; $\pm$0.03 ‰ for $\delta^{13}C$ and $\pm$0.05 ‰ $\delta^{18}O$ (GAW Report No. 242, 2017)), we concluded that the test condition did not induce any significant bias to the measurement. With respect to applying these WMO compatibility goals it should be mentioned that these precisions should be seen as the scientifically desirable level of compatibility for concurrent measurements of well-mixed background air by different laboratories, while they may not be the currently achievable best 1-$\sigma$ measurement uncertainty (GAW Report No. 242, 2017)…"

Comment #3: *This may be, but this does not in itself mean that the performance of these two systems is equal. Please elaborate a bit more - make it clearer what you want to say when mentioning this.*

Authors response: Indeed, the performance of the two Aerodyne TILDAS systems are not alike and are currently being evaluated. These performance characterizations would be described in forthcoming publications and hence, we have removed this part of the text.

- *Referee's comments on page 18:*

Authors response: All textual suggestions have been incorporated.

Comment #1: *I am missing just a bit more detail (independent from the manufacturer information) on the influence of (humid) nafion membrane on the CO2 concentration of the sampled air.*

Authors response: We have added a few more sentences, in the Discussions and conclusions section, to provide more details on the influence of the NAD on $CO_2$ mole fraction determination in samples.

"Since unbiased measurements of $CO_2$ mole fraction and its isotopic composition in whole air samples demand collection of very dry sample air, we tested and present here the results of a Nafion™ based drying system. Nafion™ dryers are an excellent alternative to chemical and recirculating chiller based dryers for mobile sampling platforms. For example, most chemical dryers either alter the chemical composition of the sample air, or are considered hazardous from a safety standpoint, especially when they are used onboard an

aircraft. On the other hand, recirculating chiller based dryers are very efficient but are large and extremely energy demanding, which makes their usage on light aircrafts logistically undesirable. Nafion™-based drying systems offer a consumable-free, reusable, and a field-deployable alternative, which does not necessitate incorporating hazardous chemicals and also eliminates the use of any power onboard an aircraft. Initial laboratory tests, using the Picarro G2301 analyser, already indicated that a Nafion™ based system did not alter the mole fraction of $CO_2$ and $CH_4$ in dry and humidified air samples and hence could potentially be a promising alternative. In this work, we tested the NAD which is configured for use with the PCP-PFP system from NOAA-ESRL, although the use of our system is not limited to that sampling platform.

…

The next requirement was to establish if the NAD was inert for the gases-of-interest and did not alter the isotopic composition of $CO_2$ while sampling. To understand the effect of the NAD on the isotopic composition of $CO_2$, we performed a semi-continuous zero-enrichment experiment with the TILDAS-SICAS instrument in our laboratory. In such an experiment, the same gas is treated both as a reference and a sample gas, where the reference stream is unprocessed and the sample stream is processed. Thus, a zero-difference between the reference and the sample stream would indicate that the processed gas was not modified at all. This is demonstrated in Figure 4, where the first part of the experiment shows that the isotopic composition of $CO_2$ is unaltered when dry sample air is passed through the NAD relative to the direct measurement of the dry sample air. The second part of the experiment demonstrates that the isotopic composition of $CO_2$, as observed when wet sample air is passed through the NAD (thus dried) relative to the direct measurement of the dry sample air, remains within the measurement uncertainties and thus indistinguishable. Since the TILDAS-SICAS is not designed for continuous measurements, we performed this experiment at a lower flow rate than what would otherwise be used in field to obtain more discrete measurements while processing a certain volume of air. This demonstrates that, even with a doubling of residence time in the NAD compared to field conditions, the isotopic composition remains unaltered. Therefore, shorter residence times during field measurements would reduce the chances of interaction between $CO_2$ and the wet membrane surface and would therefore be more favourable. Additionally, this experiment also clearly demonstrates that $CO_2$ mole fraction determinations are not significantly affected in the presence of NAD, in both dry and wet modes (sample) when compared to measurements performed without the NAD (reference). …"

- *Referee's comments on page 19:*

Authors response: All textual suggestions have been incorporated.

Comment #1-3: *please be more specific; please be more specific; I think you would do your setup more justice, if you did not limit it here to the utilisation in Brazil - even if it was constructed for this purpose. It can be used in many other settings.*

Authors response: We have added text to make this section clearer and read as follows: "We performed a storage stability check over a period of one month and the results indicated that the NAD, if stored in dry conditions i.e., filled with dry air immediately after conditioning, would perform similarly to one freshly conditioned. This was concluded by comparing the water removal capacity of the NAD and the lowest achievable water vapour concentration while processing ~200 L of humidified air (~2 %) at similar flowrates. This property is particularly beneficial for the sampling conditions in Brazil because the conditioning step is performed in the lab few days before the PFP and the NAD are shipped to the sample collection site. As such, the application of this drying unit is not only limited to sampling in Brazil, but can also be used in any other situation where drying large volumes of air samples is necessary and availability of electricity is an issue."

- *Referee's comments on page 20:*

Authors response: All textual suggestions have been incorporated.

Comment #1: *There are differences, albeit small - therefore rather, for example, the differences between the different experiments are in the range of measurement uncertainty*

Authors response: This sentence has been rephrased to: "... The second part of the experiment demonstrates that the isotopic composition of $CO_2$, as observed when wet sample air is passed through the NAD (thus dried) relative to the direct measurement of the dry sample air, remains within the measurement uncertainties and thus indistinguishable."

Comment #2: *It would make a more complete discussion if you elaborated in a bit more detail on this point (i.e. why is a shorter residence time even more favorable, what are the processes involved, surface processes, kinetic fractionation, etc.)*

Authors response: We have added the following sentence: "…This demonstrates that, even with a doubling of residence time in the NAD compared to field conditions, the isotopic composition remains unaltered. Therefore, shorter residence times during field measurements would reduce the chances of interaction between $CO_2$ and the wet membrane surface and would therefore be more favourable."

Comment #5: *I am unsure about the AMT policy this regarding, but my conviction is that all data used in a paper have to be freely available/deposited in a repository which is freely accessible, independently of the authors.*

Authors response: We now have the data freely available at https://hdl.handle.net/10411/XIDZEA.

- *Referee's comments on page 23:*

Authors response: The reference has been updated to: "Zellweger, C., Steinbrecher, R., Laurent, O., Lee, H., Kim, S., Emmenegger, L., Steinbacher, M., and Buchmann, B.: Recent advances in measurement techniques for atmospheric carbon monoxide and nitrous oxide observations, Atmospheric Measurement Techniques, 12, 5863-5878, 2019."